# A SARS-CoV-2 antibody curbs viral nucleocapsid protein-induced complement hyperactivation

Sisi Kang[1,10], Mei Yang[1,10], Suhua He[1,10], Yueming Wang[2,3,10], Xiaoxue Chen[1], Yao-Qing Chen[4], Zhongsi Hong[5], Jing Liu[6], Guanmin Jiang[7], Qiuyue Chen[1], Ziliang Zhou [1], Zhechong Zhou[1], Zhaoxia Huang[1], Xi Huang[8], Huanhuan He[1], Weihong Zheng[2,3], Hua-Xin Liao[2,3✉], Fei Xiao [1,5✉], Hong Shan[1,9✉] & Shoudeng Chen [1✉]

Although human antibodies elicited by the severe acute respiratory syndrome coronavirus 2 (SARS-CoV-2) nucleocapsid (N) protein are profoundly boosted upon infection, little is known about the function of N-reactive antibodies. Herein, we isolate and profile a panel of 32 N protein-specific monoclonal antibodies (mAbs) from a quick recovery coronavirus disease-19 (COVID-19) convalescent patient who has dominant antibody responses to the SARS-CoV-2 N protein rather than to the SARS-CoV-2 spike (S) protein. The complex structure of the N protein RNA binding domain with the highest binding affinity mAb (nCoV396) reveals changes in the epitopes and antigen's allosteric regulation. Functionally, a virus-free complement hyperactivation analysis demonstrates that nCoV396 specifically compromises the N protein-induced complement hyperactivation, which is a risk factor for the morbidity and mortality of COVID-19 patients, thus laying the foundation for the identification of functional anti-N protein mAbs.

[1] Molecular Imaging Center, Guangdong Provincial Key Laboratory of Biomedical Imaging, The Fifth Affiliated Hospital, Sun Yat-sen University, Zhuhai, China. [2] Institute of Biomedicine, Jinan University, Guangzhou, China. [3] Zhuhai Trinomab Biotechnology Co., Ltd, Zhuhai, China. [4] School of Public Health (Shenzhen), Sun Yat-sen University, Shenzhen, China. [5] Department of Infectious Disease, The Fifth Affiliated Hospital, Sun Yat-sen University, Zhuhai, China. [6] Department of Respiratory Disease, The Fifth Affiliated Hospital, Sun Yat-sen University, Zhuhai, China. [7] Department of Clinical laboratory, The Fifth Affiliated Hospital of Sun Yat-sen University, Zhuhai, China. [8] Center for Infection and Immunity, The Fifth Affiliated Hospital, Sun Yat-sen University, Zhuhai, China. [9] Department of Intervention Medicine, The Fifth Affiliated Hospital, Sun Yat-sen University, Zhuhai, China. [10] These authors contributed equally: Sisi Kang, Mei Yang, Suhua He, Yueming Wang. ✉email: tliao805@jnu.edu.cn; xiaof35@mail.sysu.edu.cn; shanhong@mail.sysu.edu.cn; chenshd5@mail.sysu.edu.cn

The fatality rate of critical condition coronavirus disease 2019 (COVID-19) patients is exceptionally high (40–49%)[1,2]. Acute respiratory failure and generalized coagulopathy are significant aspects associated with morbidity and mortality[3–5]. A subset of severe COVID-19 patients has distinct clinical features compared to classic acute respiratory distress syndrome (ARDS), with delayed onset of respiratory distress[6] and relatively well-preserved lung mechanics despite the severity of hypoxemia[7]. It has been reported that complement-mediated thrombotic microvascular injury in the lung may contribute to atypical ARDS features of COVID-19, accompanied by extensive deposition of the alternative pathway (AP) and lectin pathway (LP) complement components[8]. Indeed, complement activation is found in multiple organs of severe COVID-19 patients in several other studies[9,10], as well as in patients with severe acute respiratory syndrome (SARS)[11,12]. Patients with age-related macular degeneration (AMD, a proxy for complement activation disorders) were at significantly increased risk of adverse clinical outcomes following SARS-CoV-2 infection. Conversely, patients with complement deficiency disorders genetic background required little mechanical respiration or succumbed to their illness[13]. Together, these data suggest that hyperactive complement predispose individuals to adverse outcomes associated with SARS-CoV-2 infection.

The nucleocapsid (N) protein of severe acute respiratory syndrome coronavirus 2(SARS-CoV-2), the etiology agent of COVID-19, is one of the most abundant viral structural proteins with multiple functions inside the viral particles, the host cellular environment, and ex vivo experiments[14–20]. Among these functions, a recent preprint study found that the SARS-CoV-2 N protein bound to mannan-binding lectin (MBL)-associated serine protease 2 (MASP-2) and resulted in complement hyperactivation and aggravated inflammatory lung injury[19]. Consistently, the highly pathogenic SARS-CoV N protein was also found to bind with MAP19, an alternative product of MASP-2[21].

Although systemic activation of complement plays a pivotal role in protective immunity against pathogens, hyperactivation of complement may lead to collateral tissue injury. Thus, how to precisely regulate virus-induced dysfunctional complement activation in COVID-19 patients remains to be elucidated. The SARS-CoV-2 N protein is a highly immunopathogenic viral protein that elicits high titers of binding antibodies in humoral immune responses[22–24]. Several studies have reported the isolation of human monoclonal antibodies (mAbs) targeting the SARS-CoV-2 spike (S) protein, helping explain the possible developing therapeutic interventions for COVID-19[22,25–29]. However, little is known about the potential therapeutic applications of N protein-targeting mAbs in the convalescent B cell repertoire.

In this work, we report a human mAb (nCoV396) derived from the COVID-19 convalescent patient that specifically targets the SARS-CoV-2 N protein. The complex structure of mAb nCoV396 with N protein NTD domain reveals an allosteric regulation mechanism, which is supported with N protein-induced complement hyperactivation ex vivo assays. Our work indicates that human N-targeting mAbs from COVID-19 convalescents play essential roles in inhibition of complement hyperactivation.

## Results

**Isolation of N protein-reactive mAbs**. To profile the antibody response to the SARS-CoV-2 N protein in patients during the early recovery phase, we collected blood samples from six convalescent patients 7–25 days after the onset of the disease symptoms. All patients recovered from COVID-19 during the outbreak in Zhuhai, Guangdong Province, China, with ages ranging from 23 to 66 years. Our work and use of patients'

samples is in accordance with the declaration of Helsinki, medical ethics standards, and China's laws. Our study was approved by the Ethics Committee of The Fifth Affiliated Hospital, Sun Yat-sen University, and all patients signed informed consent forms. SARS-CoV-2 nasal swabs reverse transcription polymerase chain reaction (RT-PCR) tests were used to confirm that all six COVID-19 patients were negative for SARS-CoV-2 at the time of blood collection. Plasma samples and peripheral blood mononuclear cells (PBMCs) were isolated for serological analysis and antibody isolation. Serum antibody titers to SARS-CoV-2 S and N proteins were measured by enzyme-linked immunosorbent assay (ELISA) (Fig. 1a, b) and the specific values of antibody titers can be found in Supplementary Table 1. Serologic analysis demonstrated that serum antibody titers to the N protein were substantially higher than those to the S protein in most of the patients. For example, ZD004 and ZD006 had only minimal levels of antibody response to the S protein, while they had much higher antibody titers to the N protein. Notably, the time from disease onset to complete recovery from clinical symptoms of COVID-19 patient ZD006 was only 9 days (Supplementary Table 1).

To maximize analysis of patient ZD006 samples, who was still in the early recovery phase with a high possibility of a high percentage of antigen-specific plasma cells, single plasma cells (Fig. 1c) with the phenotype of CD3−/CD14−/CD16−/CD235a−/CD19+/CD20low-neg/CD27hi/CD38hi as well as antigen-specific memory B cells with the phenotype of CD19+/CD27+ (Fig. 1d) were sorted from PBMCs of patient ZD006 by fluorescence-activated cell sorting (FACS). To ensure an unbiased assessment, we sorted of antigen-specific memory B cells with combined probes for each recombinant protein, of which the S1 (S protein range 16-685 residues) and full-length N were separately labeled with Phycoerythrin-canin7 (PE-Cy7) and Brilliant Violet (BV421). Variable regions of immunoglobulin (Ig) heavy and light chain gene segment ($V_H$ and $V_L$, respectively) pairs from the sorted single cells were amplified by RT-PCR, sequenced, annotated and expressed as recombinant mAbs using the methods described previously[30]. All of purified antibodies were produced as IgG1 antibodies regardless their original Ig isotypes. Recombinant mAbs were screened against SARS-CoV-2 S and N proteins. In total, we identified 32 mAbs that reacted with the SARS-CoV-2 N protein, including 20 mAbs from plasma cells and 12 mAbs from memory B cells, and the Ig gene family was listed in Supplementary Table 2. We found that IgG1 is the predominant isotype at 46.9%, followed by IgG3 (25.0%), IgA (18.8%), IgG2 (6.3%), and IgM (3.1) (Fig. 1e). $V_H$ gene family usage in SARS-CoV-2 N protein-reactive antibodies was 18.8% for $V_H1$, 62.5% for $V_H3$, 9.4% for $V_H4$, 6.2% for $V_H5$ and 3.1% for $V_H7$, respectively (Fig. 1f), which was similar to the distribution of $V_H$ families collected in the NCBI database. Nine of 32 SARS-CoV-2 N protein-reactive antibodies had no mutations from their germline $V_H$ and $V_H$ gene segments (Fig. 1f and Supplementary Table 2). The average mutation frequency of the remaining mutated antibodies was 5.3% (±3.6%) in $V_H$ and 3.5% (±2.7%) in $V_L$. Various germline genes are used in N-induced antibodies from ZD006, showing strong preferences for IGHV3-30 and IGκ/λV 4-69, respectively (Supplementary Fig. 1).

Consistent with the lower serum antibody titers to the SARS-CoV-2 S protein, we identified only eight SARS-CoV-2 S protein-reactive mAbs, including five antibodies from plasma cells and three antibodies from memory B cells. The $V_H$ gene segment of the S protein-reactive antibodies had either no (6/8) or minimal (1/300) mutations (Fig. 1g). There were no significant differences in complementarity-determining region 3 (CDR3) length in amino acid residues between the N- (Fig. 1h) and S-reactive antibodies (Fig. 1i).

Approximately a quarter of antibodies directed to the N protein (Fig. 1f) and almost all of antibodies to the S protein that had no or minimal mutations from their germlines (Fig. 1g) had a

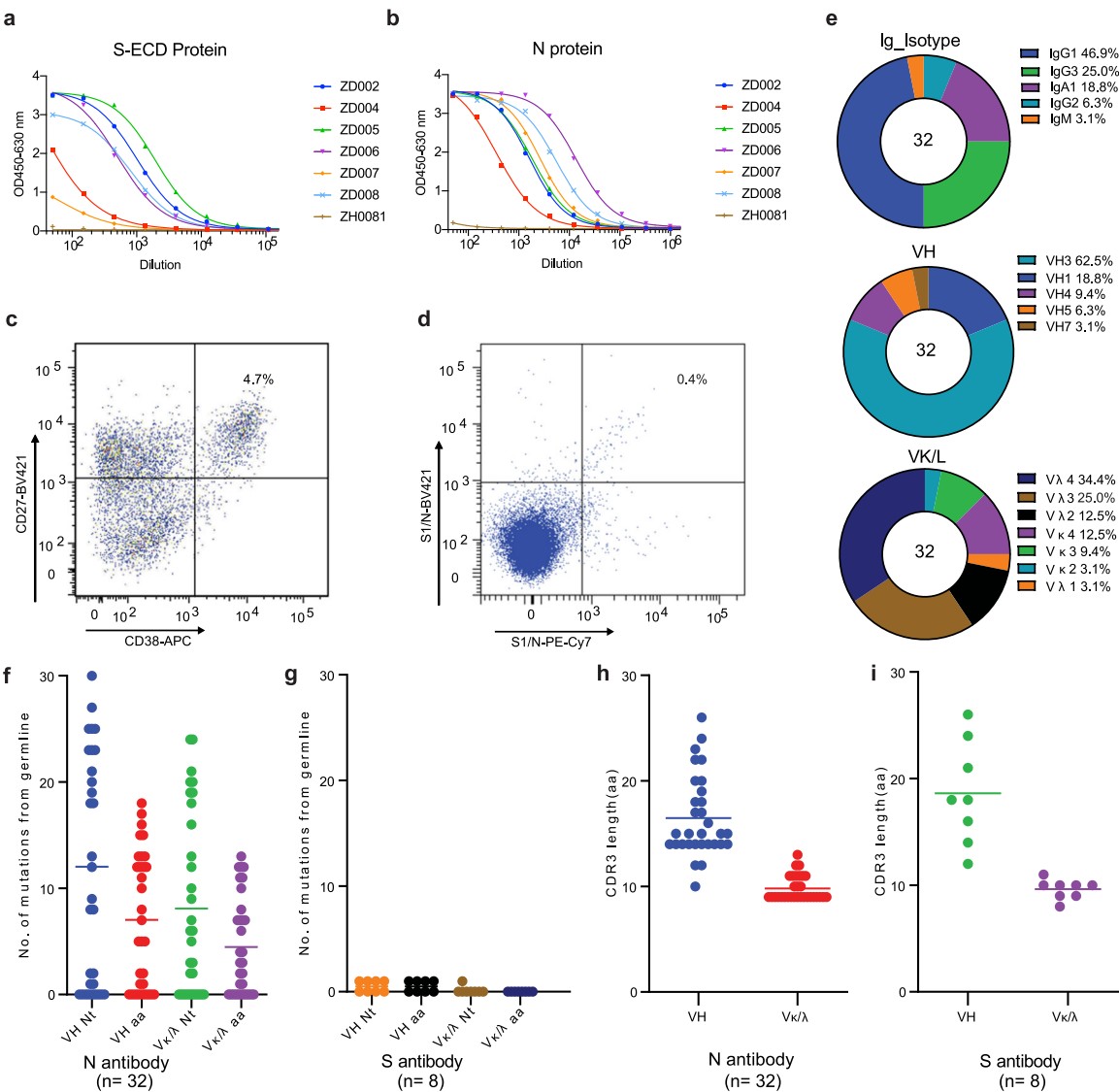

**Fig. 1 Acquisition and characterization of antibodies.** Serum antibody titers of six SARS-CoV-2 convalescent patients and a healthy person (ZH0081, non-COVID-19) to the SARS-CoV-2 S (**a**) and N (**b**) proteins measured by ELISA. All samples were performed in triplicates and mean were presented. Sorting of single plasma cells (**c**) with CD38 and CD27 double-positive B cells. **d** To minimize false positives, each of the S1 and N proteins labeled with Phycoerythrin-canin7 (PE-Cy7) and Brilliant Violet (BV421) was used to sort antigen-specific memory B cells by FACS. **e** Percentage of different isotypes, VH and VL gene families of 32 isolated N-reactive antibodies. **f** Number of mutations in nucleotides and amino acids in VH and VL (Vκ and Vλ) of 32 N-reactive antibodies and eight S-reactive antibodies (**g**). Length of the 32 N-reactive antibodies (**h**) and eight S-reactive antibodies (**i**) in H-CDR3. **f–i** Data are presented as dot and mean values.

primary antibody response similar to other typical primary viral infections. However, relatively high $V_H$ mutation frequencies (mean of 5.7%) of the majority of antibodies to the N proteins were more similar to the mutation frequencies of antibodies from the secondary responses to the influenza vaccination reported previously[31]. Although patient ZD006 was hospitalized for only 9 days after the onset of COVID-19 symptoms, the patient had high serum antibody titers, and the majority of the isolated N-reactive antibodies had a high mutation frequency, whereas the S-reactive antibodies had no or minimal mutations. These results reflect a much stronger antigen stimulation to the host driven by the SARS-CoV-2 N protein than by the S protein.

**Binding characterizations of anti-N mAbs.** To determine the antigenic targets by the N-reactive antibodies, we next analyzed the binding activities by ELISA with variant constructs of the N protein (full-length N protein (N-FL): 1-419; N protein N-terminal domain (N-NTD): 41–174; and N protein C-terminal domain (N-CTD): 250–364 (Fig. 2a). Among the 32 mAbs that bound to N-FL, 13 antibodies bound to N-NTD, and 1 antibody bound to N-CTD (Fig. 2b). A total of nine antibodies, including one antibody (nCoV400) that bound to N-CTD, six mAbs (nCoV396, nCoV416, nCoV424, nCoV425, nCoV433, and nCoV457) that bound to N-NTD and two mAbs (nCoV402 and nCoV454) that bound only to N-FL but not to the other variant N proteins, were chosen as representative antibodies for further studies. Purified antibodies were confirmed to bind the N-FL protein by ELISA (Fig. 2c). The affinity of these antibodies to the N-FL protein was measured by surface plasmon resonance (SPR; Fig. 2d). In an effort to further characterize the functional and structural relationships, three antibodies, nCoV396, nCOV416, and nCOV457, were selected for the production of recombinant

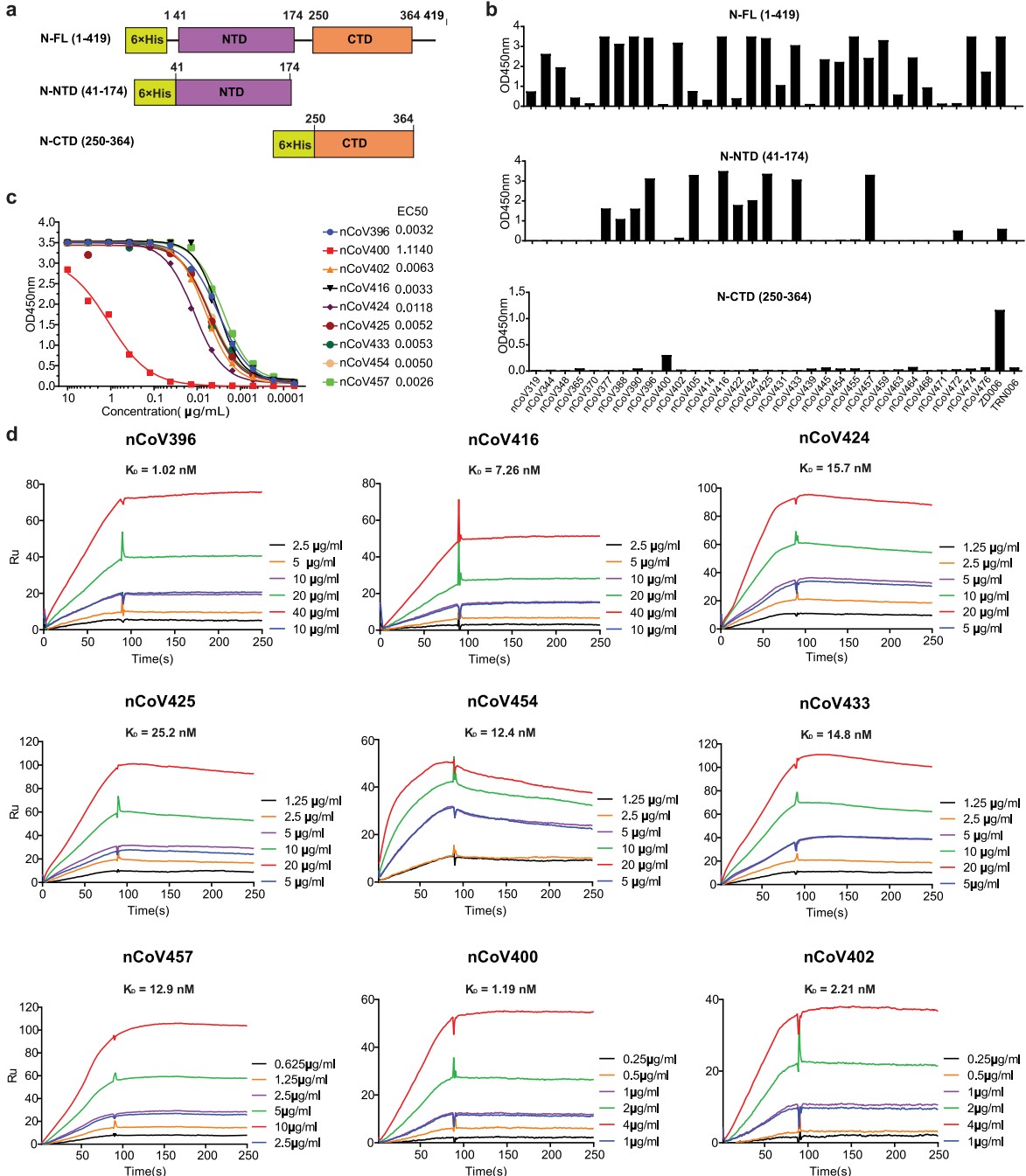

**Fig. 2 Reactivity and affinity of the isolated antibodies to the N protein antigens. a** Schematic presentation of the SARS-CoV2 N protein and two variants. **b** Antibodies expressed in transfected 293 cells were evaluated for binding to N-FL, N-NTD and N-CTD by ELISA. Plasma from the patient ZD006 and an irrelevant mAb TRN006 were used as positive control and negative control, respectively. **c** The ability of nine purified antibodies to the N-FL protein was determined by ELISA. **d** Binding affinity of nine selected antibodies to N protein were measured by SPR. KD values are shown above the individual plots.

fragment antigen-binding (Fab) antibodies based on their unique characteristics. The mAb nCoV396 has a $V_H$ mutation frequency of 2.8% but a high binding affinity with a $K_D$ of 1.02 nM (Fig. 2d) to the N protein. The mAbs nCOV416 and nCOV457 have high $V_H$ mutations at 11.1% and 8.7%, respectively and have a binding affinity to the N protein with $K_D$ values of 7.26 nM and 12.9 nM, respectively (Fig. 2d and Supplementary Table 3).

**Complex structure of mAb with N-NTD**. To investigate the molecular interaction mechanism of the mAb nCoV396 with the

N protein, we next solved the complex structure of the SARS-CoV-2 N-NTD with the nCoV396Fab at 2.1 Å resolution by X-ray crystallography. Briefly, the complex structure was determined by molecular replacement using the N-NTD structure (PDB ID: 6M3M) and monoclonal antibody omalizumab Fab (PDB ID: 6TCN) as the search models. The final structure was fitted with the visible electron density spanning residues 49–173 (SARS-CoV-2 N-NTD), 1–220 (nCoV396Fab, the heavy chain of the Fab), and 1–213 (nCoV396Fab, the light chain of Fab, except for the residues ranging from 136 to 141). For highlighting the complementary determining regions (CDRs), the Kabat nomenclature

is aligned in the Supplementary Fig. 2 as well. The complete statistics for the data collection, phasing, and refinement are presented in Supplementary Table 4.

With the help of the high-resolution structure, we were able to designate all complementarity-determining regions (CDRs) in nCoV396Fab as the following: light chain CDR1, residues 23–32 (L-CDR1), light chain CDR2, residues 51–54 (L-CDR2), light chain CDR3, residues 94–100 (L-CDR3), heavy chain CDR1, residues 26–33 (H-CDR1), heavy chain CDR2, residues 51–57 (H-CDR2), and heavy chain CDR3, residues 99–108 (H-CDR3). Among them, we identified the interaction interface between N-NTD and L-CDR1, L-CDR3, H-CDR1, H-CDR2, and H-CDR3 of nCoV396Fab with an unambiguous electron density map (Fig. 3a and Supplementary Fig. 3a).

The interacting CDRs pinch the CT tail of the SARS-CoV-2 N-NTD (residues ranging from 159 to 172), with extensive binding contacts and a buried surface area of 1079 Å$^2$ (Supplementary Table 5). Light chain L-CDR1 and L-CDR3 of nCoV396Fab interact with residues ranging from 159 to 163 of N-NTD via numerous hydrophilic and hydrophobic contacts (Fig. 3b and Supplementary Fig. 3b). Briefly, the residues G27, Y31, A32, W95, G98, I99 of variable region $V_L$ bind to 159-163 of N protein, whereas the residues I33, V50, N57, A59, E99, T100, D102, Y103, S105, S106 of variable region $V_H$ bind to 165-172 of N protein. Notably, the SARS-CoV-2 N-NTD residue Q163 is recognized by the L-CDR3 residue T95 via a hydrogen bond and simultaneously stacks with the L-CDR3 residue W96 and the L-CDR1 residue Y31 (Fig. 3c). In addition, a network of interactions from the heavy chain H-CDR2 and H-CDR3 of nCoV396Fab to residues 165-172 of N-NTD suggests that the conserved residue K169 of SARS-CoV-2 N-NTD has a critical role in nCoV396 antibody binding. K169 is recognized via hydrogen bonds with the E99 δ-carboxyl group and the T100, D102, S105 main-chain carbonyl groups inside the H-CDR3 of nCoV396Fab (Fig. 3d). In addition, SARS-CoV-2 N-NTD L167 also interacts with I33, V50, N57, and A59 of H-CDR1 and H-CDR2 of nCoV396Fab through hydrophobic interactions (Fig. 3e). Interestingly, all three residues (Q163, L167, and K169) of SARS-CoV-2 N-NTD are relatively conserved in the highly pathogenic beta-coronavirus N protein (Supplementary Fig. 4a), which implies that nCoV396 may cross-interact with the SARS-CoV N protein or the Middle East respiratory syndrome coronavirus (MERS-CoV) N protein. Indeed, the binding affinities measured by SPR analysis demonstrate that nCoV396 interacts with the SARS-CoV N protein and the MERS-CoV N protein with a $K_D$ of 7.4 nM (Supplementary Fig. 4b, c).

To discover the conformational changes between the SARS-CoV-2 N-NTD apo-state and the antibody-bound state, we next superimposed the complex structure with the N-NTD structure (PDB ID: 6M3M)[14]. The superimposition results suggest that the CT tail of SARS-CoV-2 N-NTD unfolds from the basic palm region upon nCoV396Fab binding (Fig. 3f), which likely contributes to the allosteric regulation of the normal full-length N protein function. Additionally, nCoV396Fab binding results in a 7.4-Å movement of the β-finger region outward from the RNA binding pocket, which may enlarge the RNA binding pocket of the N protein (Fig. 3f).

In summary, our crystal structural data demonstrate that the human mAb nCoV396 recognizes the SARS-CoV-2 N protein via a pinching model, resulting in a dramatic conformational change in residues 159–172, which is the linker region of N-NTD that is connected with other domains.

**MAb curbs N-induced complement activation.** Although a recent study suggests that the complement cascade is hyperactive by the N protein in the lungs of COVID-19 patients via the lectin pathway[19], it is unclear how to develop a virus-free and an effective system for analyzing the role of the SARS-CoV-2 N protein on complement hyperactivation. To this end, we developed a clinical serum-based protease enzymatic approach to assess complement activation levels in the presence of the SARS-CoV-2 N protein. Since complement activation initiated by the lectin pathway features MASP-2 proteases by specific activity for cleaving complement components 2 and 4 (C2 and C4)[32], we designed a C2 internal quenched fluorescent peptide-based analysis route for ex vivo complement hyperactivation (Fig. 4a). Briefly, the serum monitored with C3 serologic value from clinical volunteers was collected and added into the reaction system. Then we collected the fluorescence signal from cleaved C2 synthetic peptide substrates (2Abz-SLGRKIQI-Lys(Dnp)-NH$_2$) in reaction mixtures containing complement abnormal serum in the absence or presence of the SARS-CoV-2 N protein with or without the mAb nCoV396. The initial reaction rate ($V_0$) was estimated at a single concentration of individual sera from measurements over a range of substrate concentrations. The steady-state reaction constants maximal velocity ($V_{max}$) and Michaelis constant ($K_m$) were determined for comparisons (Fig. 4a).

To reach a maximal readout of the MASP-2 activity, we selected serum samples (Supplementary Table 6) with indicated C3 serologic value to analyze the serine proteases activity, as the increased C3 value for indicating activation of the complement cascade. As shown in Fig. 4b, the serum samples MASP-2 activity to C2 peptide of the healthy donors (normal C3 values, serum-01 to −03) are much weaker than those from elevated C3 serologic samples (serum-04 to −06) in the presence of SARS-CoV-2 N protein. To highlight the kinetic parameters differences in the enzymatic assays, three groups of above health serum alone with patient's serum paired data are shown in Supplementary Fig. 8a, respectively. The kinetic parameters are shown in Supplementary Table 7. These results suggest that the designed C2 internal quenched fluorescent peptide-based analysis is sensitive to viral N-induced-MASP-2 activation with abnormal C3 serologic serum samples ex vivo.

To avoid the artificial factors in serum, we next perform an in vitro fluorescent peptide-based assay without serum conditions. Only recombinant MASP-2 protein (CCP1-CCP2-SP domain, Supplementary Fig. 6) and C2 internal quenching fluorescent peptide mixed in a reaction buffer for in vitro analyzing systems. In vitro assay demonstrates that SARS-CoV-2 N protein induces hyperactivation of MASP-2 activity with the concentration gradient (Fig. 4c and Supplementary Table 8). In agreement with in vitro assays, the calculated $V_{max}$ of reactions without any other exogenous proteins is 1.49 response units (RU)·s$^{-1}$ in ex vivo reaction system (Fig. 4d). Additions of the SARS-CoV- 2N protein (concentrations ranging from 0.5 mM to 10 mM) in the reactions remarkably elevate the $V_{max}$ up to 2-fold, ranging from 2.37–3.02 RU·s$^{-1}$. Similarly, additions of the SARS-CoV-2 N protein led to an approximate 1.8-fold increase in the $V_{max}/K_m$ values, which suggested that the specificity constant ($K_{cat}/K_m$) of MASP-2 to substrates is increased in the presence of the viral N protein as the enzyme concentrations are equivalent among the reactions (Fig. 4d and Supplementary Table 9). Taken together, our works reveal that these systems truly representative of MASP-2 protease-mediated cleavage of C2. To confirm the kinetic analyses, Hanes plots ([S]/V versus [S]) were also drawn and found to be linear (Fig. 4e). Therefore, the addition of the SARS-CoV-2 N protein does not change the single substrate binding site characterization of the enzymatic reactions. To assess the suppression ability of nCoV396 to the SARS-CoV-2 N protein-induced complement hyperactivation function, we next conducted complement hyperactivation analyses at various N protein: nCoV396

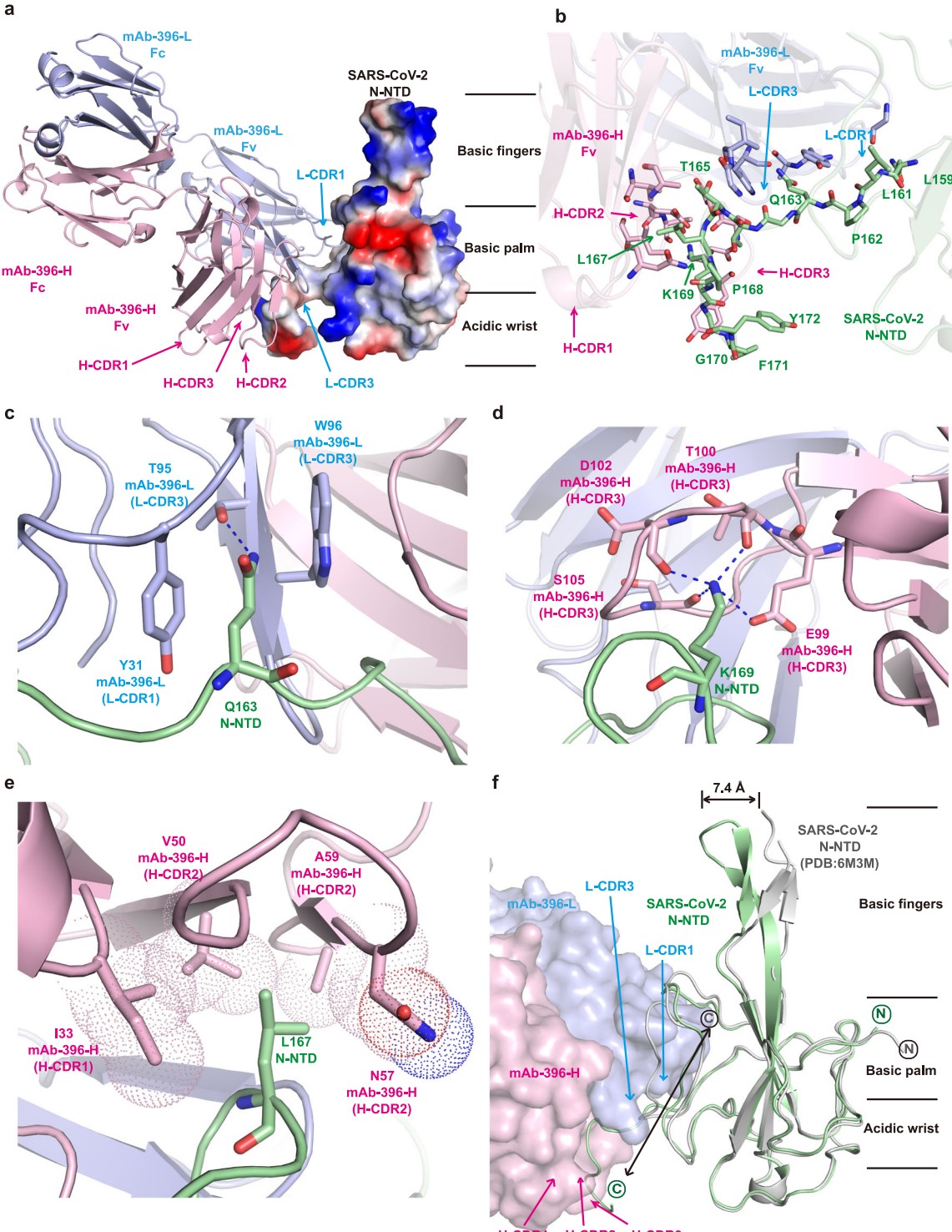

**Fig. 3 Complex structure of mAb nCoV396 with SARS-CoV-2 N-NTD. a** Overall structure of the mAb nCoV396-SARS-CoV-2 N-NTD complex. The light chain (pink) and heavy chain (blue) of mAb nCoV396 are illustrated with the ribbon representation. SARS-CoV-2 N-NTD is illustrated with electrostatics surface, in which blue denotes a positive charge potential while red indicates a negative charge potential. **b** The N-NTD epitope recognized by mAb nCoV396. The interacting residues of N-NTD and nCoV396 are highlighted with the stick representation. Recognition of Q163 (**c**), K169 (**d**), and L167 (**e**) in N-NTD by mAb nCoV396. The dashed blue line represents hydrogen bonds. Hydrophobic interactions are illustrated with the dot representation. **f** Conformational changes of N-NTD upon mAb nCoV396 binding. The apo structure of N-NTD is colored with gray. Antibody-bound N-NTD is colored green. The N-terminus and C-terminus of the N-NTD are labeled with circles. mAb nCoV396 is illustrated with surface representation. All figures were prepared by Pymol.

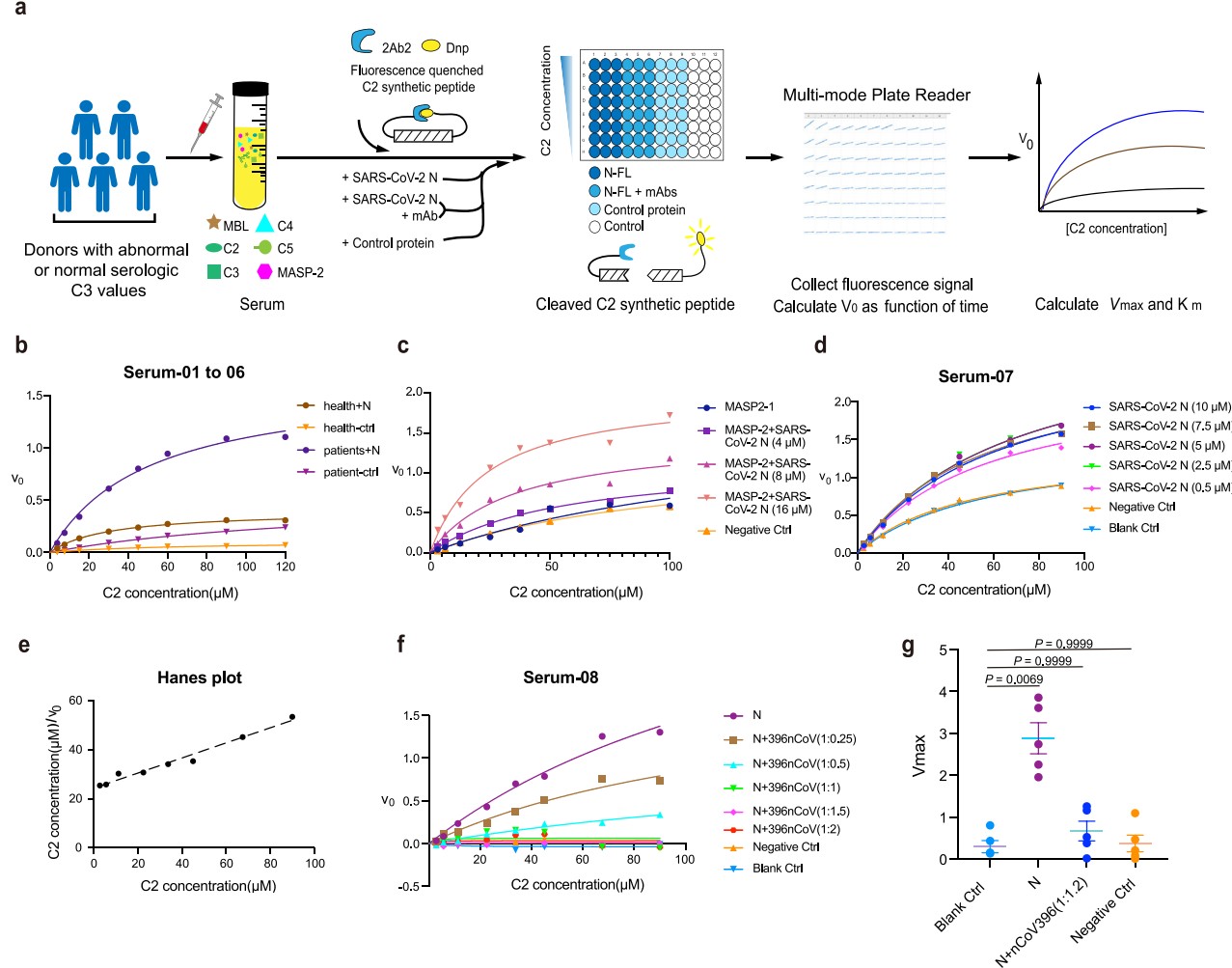

**Fig. 4 Antibody nCoV396 compromises SARS-CoV-2 N protein-induced complement hyperactivation. a** Flow scheme of the SARS-CoV-2 N protein and nCoV396 influencing the protease activity of MASP-2 in the serum from donors. **b** Serum-01 to 06 are used to compare the MASP-2 activity to C2 of serum sample with normal C3 (Health-110,113,117, $n = 3$) and serum sample with abnormal C3 (Patient-81,123,130, $n = 3$), and the Michaelis–Menten curves of are presented as mean (three groups of above health serum alone with patient's serum paired data are shown in Supplementary Fig. 8a). **c** The Michaelis–Menten curve of N protein-induced excessive cleavage of C2 in the presence of recombinant MASP-2 in vitro. The reaction system without N protein, the increase of N protein concentration and negative control protein (ENL) expressed in *E. coli* are presenting. The Michaelis–Menten curve shows the effect of increasing the N protein concentration (**d**) and antibody concentration (**f**) on the substrate C2 cleavage of MAPS-2 in the Serum-07 and Serum-08. **e** A Hanes plot where C2 concentration/V0 is plotted against C2 concentration with the addition of 5 μM N protein. **b**–**d**, **f** All samples were performed in triplicates and mean were presented. **g** Five serum sample from biologically independent donors ($n = 5$) with abnormal serologic C3 values. (Serum-08 to −12). And we used Michaelis–Menten equation to calculate the $V_{max}$ (with experimental data from Fig. 4f (Serum-08), Supplementary Fig. 5b (Serum-09 to −11), and Supplementary Fig. 7a (Serum-12)). Each sample was performed in triplicates and mean values ± SEM of $V_{max}$ are presented. Two-sided Kruskal–Wallis test with Dunnett's multiple comparisons test was used for comparing the $V_{max}$ of groups. The significant reference is 0.05.

ratios. As shown in Fig. 4f, the addition of the N protein elevates the $V_{max}$ value up to 40-fold, whereas the addition of the antibody nCoV396 decreases the $V_{max}$ in a dose-dependent manner (Supplementary Table 10). To further validate the function of nCoV396, we next performed complement hyperactivation analyses in other serum samples with abnormal serologic C3 values. Consistently, the $V_{max}$ of reactions was boosted in the presence of the N protein in all samples but declined in the presence of both the mAb nCoV396 and the N protein (Fig. 4g). The kinetic enzyme parameters are listed (Supplementary Table 11 and Supplementary Fig. 7b) and the Michaelis–Menten curves are presented in Supplementary Figs. 5b and 7a (Serum 08-12). To validate that any other N-specific mAbs have similar functions as nCoV396, we next sought to identify the linear epitope of the mAbs to N protein. Briefly, 68 overlapping 18-mer peptides derived from the full-length

N-protein have been synthesized and used for epitope screening. The binding capacity of the mAb candidate to the epitopes was analyzed by enzyme-linked immunosorbent assay (ELISA). Consistently with nCoV396, several mAbs (nCoV454, nCoV457, and nCoV416) display positivity against the peptide epitopes of N protein located between 157 and 180 residues (Supplementary Table 12). The data indicates that the 162–170 region of SARS-CoV-2 N protein is a vital epitope. Next, we verified the functions of these two antibodies (nCoV454 and nCoV457) in the complement hyperactivation ex vivo assays. As shown in Supplementary Fig. 7a, the $V_{max}$ of N-induced complement hyperactivation (plum curve) are dramatically reduced in the presence of nCoV454 (black curve) and nCoV457 (brown curve), as well as nCoV396 (dark blue curve; Supplementary Fig. 7a, b). To further validate whether these mAbs work against other highly pathogenic relative N proteins in the

ex vivo complement system, we next perform the ex vivo assays in the presence of SARS-CoV N-protein. As shown in Supplementary Fig. 7c, the SARS-CoV-2 N-directed mAbs (nCoV396, nCoV454, and nCoV457) display potent inhibition to SARS-CoV N protein-induced MASP-2 hyperactivation with decreased $V_{max}$ in the assays (Supplementary Fig. 7d). Furthermore, the addition of mAb nCoV396 disrupts the N protein binding ability for transcriptional regulatory sequence (TRS) (Supplementary Fig. 8). Taken together, our data support N protein's similar mechanisms in SARS-CoV, and the mAbs isolated in this work function as effective inhibitors for N protein-induced complement hyperactivation.

## Discussion

From a quickly recovered COVID-19 patient, we isolated 32 mAbs specifically targeting the SARS-CoV-2 N protein. The binding affinity of mAbs ranged from 1 nM to 25 nM, which is comparable with the binding affinity of mature S protein-reactive antibodies[22,25–29] and the other mature antibodies identified during acute infections[33,34]. The characteristics of the isolated N-reactive mAbs are different from those of the isolated S-reactive mAbs in COVID-19 patients during the early recovery phase, suggesting that sampling time is pivotal for identifying differential immune responses to different SARS-CoV-2 viral proteins. Due to the small sample size, it cannot be said that this is a common phenomenon in the human population. Indeed, McAndrews et al.[35] reported that the detection of N-specific antibodies does not always correlate with the presence of S-RBD neutralizing antibodies. Consistently with our observation, Hachim et al.[36] have reported that antibodies to N protein developed earlier than S protein-specific antibodies. Similarly, Sun et al.[17] found that N-IgG was significantly higher in ICU patients than in non-ICU patients.

The crystal structure of nCoV396 bound to SARS-CoV-2 N-NTD elucidates the interaction mechanism of the complex between the first reported N protein- reactive human mAb and its targeted N protein. Three conserved amino acids (Q163, L167, and K169) in the N protein are responsible for nCoV396 recognition, which provides evidence of cross-reactivity of nCoV396 to the N protein of SARS-CoV or MERS-CoV. Intriguingly, the nCoV396 binding of SARS-CoV-2 N-NTD undergoes several conformational changes, resulting in an enlargement of the N-NTD RNA binding pocket enlargement and partial unfolding of the basic palm region. More importantly, this conformational change occurs in the CT tail of the N-NTD, which may alter the positioning of individual domains in the context of the full-length protein and lead to a potential allosteric effect for protein functions.

Complement is one of the first lines of defense in innate immunity and is essential for cellular integrity and tissue homeostasis and for modifying the adaptive immune response[37]. Emerging evidence suggests that the complement system plays a vital role in a subset of critical COVID-19 patients, with features of atypical acute respiratory distress syndrome, disseminated intravascular coagulation, and multiple organ failure[9,10,38]. A few pieces of evidence show that the N protein of highly pathogenic coronaviruses (SARS-CoV-2 and SARS-CoV) is involved in the initiation of MASP-2-dependent complement activation[19,21]. Encouragingly, critical COVID-19 patients treated with complement inhibitors, including small molecules to the complement component C3 and an antibody targeting the complement component C5, show remarkable therapeutic outcomes[19]. Currently, there are 11 clinical trials related to targeting the complement pathway (https://clinicaltrials.gov). To avoid adverse effects of human complement component-targeting therapy, a viral protein-specific approach is warranted. The antibody nCoV396 isolated

from COVID-19 convalescent patient is an excellent potential candidate with a high binding affinity to the N protein and high potency to inhibit complement hyperactivation. As revealed by atomic structural information, the binding may allosterically change the full-length N protein conformation. To determine the role of nCoV396 in the suppression of complement hyperactivation, we monitored MASP-2 protease activity based on its specific fluorescence-quenched C2 substrate in sera from patients with abnormal serologic C3 values. The active complement components in the sera of patients with elevated C3 value allow us to monitor the activating effects of the SARS-CoV-2 N protein and its specific mAbs. Although we cannot calculate the other steady-state enzymatic reaction constants as the precise concentration of MASP-2 in serum is unknown, we identified the $V_{max}$ of the specific C2 substrate for the enzymatic reaction. We demonstrated that the SARS-CoV-2 N protein elevated the $V_{max}$ of the reaction, up to 40-fold, in the sera of all 7 individuals tested, while nCoV396 effectively suppressed the $V_{max}$ of the reaction mixtures. Additionally, three other N-specific mAbs nCoV454, nCoV457- and nCoV416, which target to similar epitopes of the N protein, display effective inhibitions in the ex vivo serum-based complement activation assays. These results indicated that serum-based complement activation analysis is a virus-free and an effective method for examining complement activation mediated by the SARS-CoV-2 N protein. Consistently, the subsequent in vitro fluorescent assays reveal that N-specific mAbs function as effective inhibitors for N protein-induced complement hyperactivation.

Although the precise interaction of the SARS-CoV-2 N protein with MASP-2 remains to be elucidated, our work defined the region on the SARS-CoV-2 N protein recognized by the mAb nCoV396 that plays an important role in complement hyperactivation and indicates that human mAbs from convalescents could be a promising potential therapeutic candidate for the treatment of COVID-19.

For the other 18 antibodies, however, it is difficulty to express the rest of the nucleocapsid portion separately due to the nature of the protein. These regions belong to disorder or flexible parts of the protein, although we have worked on these several times. The comprehensive studies have suggested that the nucleocapsid compact functional domains are its NTD and CTD. These two domains play several vital roles in viral RNA recognition, viral genomic RNA packing, high-order structure formation of viral ribonucleoproteins (RNP), etc. Therefore, we focus our subsequent studies on the monoclonal antibodies that bind to N-NTD, N-CTD, or full-length protein in this project. Nevertheless, we believe that it is worth checking whether these monoclonal antibodies are bound to other regions of the nucleocapsid protein in the future work.

## Methods

**Recombinant SARS-CoV-2 S-ECD and N proteins**. Recombinant SARS-CoV-2 S protein (extracellular domain of the S protein (ECD) with His and FLAG Tags, Z03481) was purchased from GenScript. Recombinant SARS-CoV-2 full-length N protein with a CT 6x His tag (His tag, 40588-V08B) was purchased from Sino Biological. The SARS-CoV-2 N protein expression plasmid pET-28a (SARS-CoV-2 N-FL) was a gift from the Guangdong Medical Laboratory Animal Center. SARS-CoV and MERS-CoV N-FL expression plasmid (pET-28a) were purchased from RuiBiotech. SARS-CoV-2 N-NTD domain (residues 41–174) and SARS-CoV-2 N-CTD domain (residues 250–364) were cloned into the pET-28a vector by PCR and the primers used are shown in Supplementary Table 13. Expression of SARS-CoV, MERS-CoV N-FL, SARS-CoV-2 N-FL and variants in Rosetta *E. coli* was induced with 0.1 mM isopropylthio-β-galactoside (IPTG) and cultured overnight at 16 °C in Terrific Broth media. Expressed recombinant N proteins were initially purified by using nickel column chromatography and further purified via size-exclusion chromatography.

**Ethical approval**. The study was conducted according to the declaration of Helsinki and China's laws. It was approved by Ethics Committee of the Fifth Affiliated

Hospital of Sun Yat-sen University (No. K198-1). Written informed consent was obtained from all study participants.

**Sorting of single plasma cells and memory B cells by FACS**. Blood samples were collected 9–25 days after the onset of the disease from patients who had recovered from COVID-19 infection. PBMCs and plasma were isolated from blood samples by Ficoll-Paque PLUS (GE, 17-1440-02) density gradient centrifugation. Single plasma cells with the surface markers CD3− (1:50 dilution), CD14− (1:200 dilution), CD16− (1:50 dilution), CD235a− (1:200 dilution), CD19+ (1:50 dilution), CD20low-neg (1:40 dilution), CD27hi (1:50 dilution), IgD− (1:40 dilution), and CD38hi (1:10 dilution) and memory B cells with the surface markers CD3− (1:50 dilution), CD14− (1:10 dilution), CD16− (1:50 dilution), CD235a− (1:200 dilution), CD20− (1:40 dilution), CD19+ (1:50 dilution), CD27+ (1:50 dilution), and IgD− (1:40 dilution) were sorted by fluorescence-activated cell sorting (FACS). To minimize false positives in the sorting of antigen-specific memory B cells, streptavidin was labeled separately with Phycoerythrin-canin7 (PE-Cy7) and Brilliant Violet 421. Labeling with each fluorophore was carried out on separate aliquots of streptavidin, which were then mixed together prior to interaction with biotinylated the S1 and N proteins used for sorting. Cells showing elevated fluorescence for both PE-Cy7- and BV421-labeled S or N protein were sorted into single well of 96-well plates containing cell lysis and RT buffer for Ig gene amplification by FACS[39] on a BD FACS Aria SORP. Data were analyzed using BD FACS Diva 8.0.1 software.

**Isolation and expression of Ig $V_H$ and $V_L$ genes**. Genes encoding $V_H$ and $V_L$ were amplified by reverse transcription (RT) and nested primer chain reaction (PCR) and nested PCR[30]. PCR products of Ig $V_H$ and $V_L$ genes were purified using a PCR purification kit (QIAGEN), sequenced in forward and reverse directions (Thermo Fisher scientific) and annotated by using IMGT/V-QUEST (www.imgt.org/IMGT_vquest). Functional $V_H$ and $V_L$ genes were used for assembling full-length Ig heavy and light chain linear expression cassettes by one-step overlapping PCR[30]. HEK-293T in 12-well plates were transfected with the assembled Ig heavy and light chain pairs derived from the same single individual plasma cells using Effectene (QIAGEN) as the transfection reagent[30].

**Production of recombinant IgG and Fab antibodies**. For the production of purified full-length IgG1 antibodies, the $V_H$ and $V_L$ genes were cloned into the pCDNA3.1+ (Invitrogen)mammalian expression vector containing either the human *IgG1* constant region gene, the human kappa light chain constant region gene or the lambda light chain constant region gene using standard recombinant DNA technology[30]. For the production of the purified nCoV396Fab antibody, a stop codon TGA was introduced after the sequence (5′-TCTTGTGACAAA-3′), which encodes the amino acid residues SCDK, just before the hinge of the human IgG1 constant region[40]. Recombinant IgG1 antibodies and the nCoV396 antibody Fab fragments were produced in 293F cells cultured in serum-free medium by co-transfection with the generated IgG1 full-length or Fab heavy- and light chain gene expression plasmid pairs using polyethylenimine[41]. Full-length IgG1 antibodies were purified by using Protein A column chromatography as described previously[30]. The nCoV396 antibody Fab used for the crystal structure was purified by Lambda FabSelect, an affinity resin designed for the purification of human Fab with a lambda light chain (GE Healthcare)[40].

**Analysis of mAbs to the S and N proteins by ELISA**. We collected plasma from six patients and measured serum antibody titers using recombinant SARS-CoV-2 S and N proteins as antigens to coat ELISA plates. Antibodies in the supernatant of the transfected 293T cultures harvested 3 days after transfection were screened by ELISA[30]. The binding of purified antibodies to the N or S protein was also analyzed by ELISA. Briefly, all protein antigens (S, N-FL, N-NTD (41–174aa), and N-CTD (250-364aa)) were used at 200 ng/well to coat 96-well high-binding ELISA plates (Nunc 442404) using carbonate-bicarbonate buffer at pH 9.6. Plates were incubated overnight at 4 °C and blocked at room temperature for 2 h with PBS blocking buffer containing 5% w/v goat serum and 0.05% Tween 20. Plasma or supernatant of transfected 293T cell cultures or purified mAbs in serial dilutions in PBS were incubated at 37 °C for 1 h. Goat anti-human IgG-horseradish peroxidase (HRP, 1:10,000 dilution; Promega, W4031) as the secondary antibody diluted in blocking buffer was added and incubated at 37 °C for 1 h. These plates were then washed five times with PBS and developed with 100 μL of 3,3′,5,5′-Tetramethylbenzidine (TMB) substrate/well (Solarbio PR1200). The reaction of the plates was stopped with 50 mL of 2 M $H_2SO_4$/well and read at a wavelength of 450 nm by an ELISA reader. The relative affinity of mAbs to the N protein antigen was determined as the effective concentration (EC$_{50}$) of the antibody resulting in half maximal binding to the antigen by curve fitting with GraphPad Prism Version 8.0 (GraphPad Software, San Diego, CA).

**Affinity and kinetic measurements by SPR**. The binding affinity ($K_D$), association rate ($K_a$), and dissociation rate ($K_d$) of purified mAbs to the N protein were determined by SPR using a Biacore X100 System. Anti-human Fc IgG antibody was first immobilized on a CM5-chip to ~6000 RU by covalent amine coupling using a human antibody capture kit (GE Healthcare). Purified mAbs were captured on channel 2 of the CM5 chip to ~200 RU. Five 2-fold serial dilutions of the N protein starting at

40 mg/ml, 20 mg/ml, 10 mg/ml, or 4 mg/ml were injected at a rate of 30 mL/min for 90 s with a 600-s dissociation. The chip was regenerated by injection of 3 M MgCl$_2$ for 30 s. All experiments were performed at room temperature, and data were analyzed using Biacore X100 Evaluation Software (version: 2.0.1). Curves were fitted to a 1:1 binding model to determine the kinetic rate constants ($K_a$ and $K_d$). $K_D$ values were calculated from these rate constants.

**Crystallization and data collection**. SARS-CoV-2 N-NTD (41–174aa) was cloned into the pRSF-Duet-1 vector with an NT 6x His-SUMO tag, recombinantly expressed in Rosetta *E. coli* and purified as an NT 6x His-Sumo-tagged protein. After Ni column chromatography followed by ulp1 digestion for tag removal, the SARS-CoV-2 N-NTD (41–174) protein was further purified via size-exclusion chromatography. Prior to crystallization, the SARS- CoV-2 N-NTD (41–174) sample was mixed with nCoV396Fab at a 1:1.5 molar ratio for approximately half an hour and then further purified via size-exclusion chromatography. Crystals were grown by the sitting drop method using a Mosquito LCP crystallization robot with 0.3 μL of protein (6 mg/ml) mixed with 0.3 μL of well solution at 16 °C. Better crystals were obtained in 0.01 M calcium chloride dihydrate, 0.05 M sodium cacodylate trihydrate (pH 7.2), 1.675 M ammonium sulfate, and 0.5 mM spermine. The crystals were harvested after 3 days. Crystals were frozen in liquid nitrogen in the reservoir solution supplemented with 25% glycerol (v/v) as a cryoprotectant. X-ray diffraction data were collected at the Shanghai Synchrotron Radiation Facility BL18U at a wavelength of 0.979 Å and a temperature of 100 K. The X-ray data were indexed and integrated using the program HKL3000 v716.1. Phenix-1.17.1-3660. The complex structure of SARS-CoV-2 N-NTD with the mAb nCoV396 was determined by Phenix-1.17.1-3660 molecular replacement using the SARS-CoV-2 N-NTD structure (PDB ID: 6M3M) and the monoclonal antibody omalizumab Fab (PDB ID: 6TCN) as the search models. The model building was carried out manually by using the program Coot 0.9.3 EL. All figures were prepared by PyMOL 0.99. The X-ray diffraction and structure refinement statistics are summarized in Supplementary Table 4. The final Ramachandran statistics are 97.1% favored, 2.9% allowed and 0.0% outliers.

**Fluorescence-quenched substrate assays**. MASP-2 cleaves the complement components C4 and C2 to form the C3 convertase C4b2a. The fluorescent peptide-based assay was developed using a C2-derived peptide sequence (SLGRKIQI) conjugated to Dnp fluorescent group. The synthetic fluorescent peptides were quenched by their N-terminal 2Abz group. Once cleavage occurs, which is mediated explicitly by MASP-2, the fluorescent Dnp group is released. This release can be monitored over a time course by a spectrofluorometer using an excitation wavelength of 320 nm and an emission wavelength of 420 nm. The fluorescence-quenched substrate (FQS) [C2 P4-P4′ (2Abz-SLGRKIQI-Lys(Dnp)-NH$_2$)] was synthesized and purified in greater than 90% purity by Sangon Biotech (Shanghai, China). The FQS was solubilized in 50% (v/v) dimethylformamide (DMF). Ex vivo assays were carried out in fluorescence assay buffer (FAB) (0.05 M Tris-HCl, 0.15 M NaCl, 0.2% (w/v) PEG 8000, and pH 7.4) at 37 °C. In total, 100 μL/well reaction consists of 2.5 μL serum from patients with abnormal serologic C3 values, 47.5 μL final concentration of 5 mM SARS-CoV-2 N protein, 47.5 μL final concentration of SARS-CoV-2 N protein (5 mM) mixed with antibodies (6 mM), or 47.5 μL same concentration of BSA or other proteins expressed in *E. coli*, 50 μL fluorescent substrates with final substrate concentrations in the range of 2.8125–90 μM or 3.75–120 μM in triplicate. The plates were incubated at 37 °C for several minutes. The serum samples were detected by Clinical Laboratory Group of Department of Experimental Medicine of The Fifth Affiliated Hospital, Sun Yat-sen University. Our work and use of patients' samples is in accordance with the declaration of Helsinki, medical ethics standards and China's laws. Our study was approved by the Ethics Committee of The Fifth Affiliated Hospital, Sun Yat-sen University, and all patients signed informed consent forms. In addition, for adolescent patients, their legal guardian signed informed consent forms. To avoid the artificial factors in serum, we next perform an in vitro fluorescent peptide-based assay without serum conditions. 100 μL/well reaction consists of 7.5 μL 320 nM recombinant MASP-2 protein (CCP1 + CCP2 + SP domain), 42.5 μL final concentration of 4 μM, 8 μM or 16 μM SARS-CoV-2 N protein, or 42.5 μL same concentration of other proteins expressed in *E. coli*, 50 μL fluorescent substrates with final substrate concentrations in the range of 3.125–100 μM in triplicate. The plates were incubated at 37 °C for several minutes.

Fluorescence intensity was measured on an EnVision 2015 Multimode plate reader (PerkinElmer) at an excitation wavelength of 320 nm and an emission wavelength of 420 nm for the FQS. The initial reaction rate was estimated at a single concentration of enzyme from duplicate measurements over the range of substrate concentrations (at least 3000 s). To determine steady-state reaction constants (Km, $k_{cat}$, $k_{cat}/K_m$, $V_{max}$, $K_{0.5}$, and $h$ [Hill coefficient]), the experimental results were fitted, using GraphPad Prism Version 8.0 (GraphPad Software, San Diego, CA) to the Michaelis-Menten single site binding equation $V = V\max \times (\frac{C}{C+Km})$, which explains the relationship between reaction rate and substrate concentration, or an equation describing positive cooperativity $V = V_{max} \times C/(C + K_{0.5})$, which defines the relationship between reaction rate and substrate concentration when more than one binding site applies. Hanes plots ([S]/V versus [S]) were also drawn to confirm the kinetic analyses. The catalytic efficiency ($K_{cat}$) values were calculated as $K_{cat} = V_{max}/[E]$ Plots of residual values for fits to the Michaelis–Menten equation or the

equation for positive cooperativity were carried out to further resolve the best fitting of the data.

**Expression and purification of recombinant MASP-2 (287–686aa).** The recombinant MASP-2 (287-686aa) fragment had constructed into a pRSF-Duet-1 vector with N-terminal His-SUMO tag (the primers used are shown in Supplementary Table 13) and expressed in *E. coli*. strain Rosetta. MASP-2 (287–686aa) fragment was induced with 1 mmol/L isopropylthio-β-galactoside (IPTG) and incubated 5 h at 37 °C in the Terrific Broth media. The cells were collected in 30 mL of 50 mM Tris-HCl, 20 mM EDTA, pH 7.4. The cells were thawed and sonicated on ice for 6 × 30 s, with at least 10 s interval. The inclusion bodies were collected by centrifugation (27,000×*g*, 50 mins, 4 °C) and the supernatant was discarded. The pellet was washed three times with 10 mL of 50 mM Tris-HCl, 20 mM EDTA, pH 7.4. The inclusion bodies were solubilized in 6 M guanidine hydrochloride (Gu-HCl), 0.1 M Tris-HCl, 100 mM DTT, pH 8.3, at room temperature (RT) for 3 h, and then denatured inclusion bodies were dialyzed the into 6 M urea. After nickel column chromatography, the purified proteins were diluted at 4 °C overnight into the refolding buffers (50 mM Tris-HCl, 3 mM reduced glutathione, 1 mM oxidized glutathione, 5 mM EDTA, and 0.5 M arginine, pH 9.0). Concentrate of the refolding protein were further purified using a Superdex 75 16/60 column in PBS, aliquoted, snap frozen and maintained at −80 °C.

**Linear epitope mapping with peptide-ELISA assay.** The 18-mer C-terminal biotinylated synthetic peptides, spanning the full length of SARS-CoV-2 N protein, were synthesized by Sangon Biotech (Shanghai). The reactivity of synthetic peptides with mAb nCoV396, nCoV454, nCoV457, and nCoV416 were measured by enzyme-linked immunosorbent assay (ELISA). Briefly, 96-well plates were coated with 100 μL/well of individual peptide (10 μg/mL in Phosphate-Buffered Saline (PBS) buffer) at 4 °C overnight. An unrelated mAb TT017 were used as a negative controls and streptavidin as a positive control. The wells were then incubated sequentially with 100 μL/well of 1x phosphate-buffered saline, 0.1% Tween 20 Detergent (PBST) plus 5% skimmed milk powder at 37 °C for 1 h, and 1 μg of mAb at 10 μg/mL was incubated at 37 °C for 1 h. Goat anti-human IgG-horseradish peroxidase (HRP, 1:10,000 dilution; Promega, W4031) in PBST plus 1% milk powder was used as secondary antibody at 37 °C for 1 h. Five washes with PBST were carried out between incubation steps. For color development, 100 μL/well of 3,3′,5,5′-tetramethylbenzidine (TMB) mixture was added and incubated for 10 min, followed by addition of 50 μL/well of 1 M $H_3PO_4$ to stop the reaction. Absorbance was measured at 450 nm in a 96-well plate reader.

**Electrophoretic mobility shift assay of SARS-CoV-2 N protein with viral transcriptional regulatory sequence.** All experiments were conducted in EMSA buffer (10 mM $Na_2HP_3O_4$ and $NaH_2P_3O_4$, 150 mM KCl, 5% glycerol with biotin-labeled transcriptional regulatory sequence (TRS) RNA: Biotinylated-5′-AAGUU CGUUU-3′ (Sangon Biotech, Shanghai). Reactions were set up in 20 μL aliquots each containing 5 nM biotinylated-labeled TRS RNA. SARS-CoV-2 N protein or with nCoV396 mAb were added to the aliquots starting at a concentration of 32 μM, with each following aliquot containing a 2-fold serial dilution of the proteins. A control reaction was set up where only TRS RNA or only the highest concentration of protein and buffer were added. The aliquots were allowed to react at 25 °C for 30 mins, and then were loaded on a 1x TBE buffer RNA retardation gel. The gel was run at 80 V and 4 °C for 2 h. The RNAs were subsequently transferred to Hybond N⁺ membrane (GE Healthcare) under 100 V for 15 min at 4 °C before visualized the Chemiluminescent Nucleic Acid Detection Module Kit (Thermo Scientific). Quantitation of the free TRS RNA band was achieved through the ImageJ software 2.0.0-rc-69/1.52p (National Institutes of Health, Bethesda, MA). Bound TRS RNA was estimated by subtracting the free TRS RNA band of each reaction from that of the control lane. The fraction of bound TRS RNA was fitted against the equation: $Y = \frac{1}{1+\left(\frac{Kd}{X}\right)n}$ using GraphPad Prism Version 8.0 (GraphPad Software, San Diego, CA), where $Y$ is the fraction of TRS RNA bound to the protein, $X$ is the protein concentration, $K_d$ is the dissociation constant.

**Reporting summary**. Further information on research design is available in the Nature Research Reporting Summary linked to this article.

## Data availability

Data generated or analyzed during this study are included in this published article (and its supplementary information files). Source data are provided with this paper. Any other raw data pertaining to this study are available from the corresponding author upon reasonable request. The coordinates and structure factors files for the complex of SARS-CoV-2 N-NTD with nCoV396 was deposited to Protein Data Bank with accession code 7CR5. Source data are provided with this paper.

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

## Acknowledgements

This work is supported by the COVID-19 Emerging Prevention Products, Research Special Fund of Zhuhai City (ZH22036302200016PWC to S.C.; ZH22036302200028PWC to F.X.; ZH22046301200011PWC to H-X.L.); National Natural Science Foundation of China (No. 31770801) to S.C.; Natural Science Foundation of Guangdong Province, China (No. 2018B030306029) to S.C.; Emergency Fund from Key Realm R&D Program of Guangdong Province (2020B111113001) to H.S.; Zhuhai Innovative and Entrepreneurial Research Team Program (ZH01110405160015PWC, ZH01110405180040PWC) to H-X.L. We thank the staffs of the BL18U/19U/17U beamlines at SSRF for their help with the X-ray diffraction data screening and collections. We thank Junlang Liang, Tong Liu, Nan Li, Xiaoli Wang, Zhenxing Jia, and Jiaqi Li from Zhuhai Trinomab Biotechnology Co., Ltd. for technical assistants of mAbs isolation, production, and characterization.

## Author contributions

S.C., H.S., F.X. and H-X.L. contributed the conception of the study and established the construction of the article. S.C. and H-X.L. designed the experiments and wrote the manuscript. S.K., M.Y. and S.H. contributed to protein purification and crystallization, in vitro protein–protein interaction analysis, and complement activation analysis. Y.W. contributed to mAbs isolation, in vitro protein–protein interaction analysis. S.C., S.K., M.Y. and S.H. performed structural determination and validation. S.C., S.K. and Y.W. drew figures. X.C., Y.C., Q.C., Ziliang Z., Zhechong Z., Zhaoxia H., X.H., H.S., W.Z. and H.H. contributed to interpretation of data. Zhongsi H., J.L., G.J. and F.X. contributed to clinical samples collections. S.K., M.Y. S.H. and Y.W. contributed equally to this work.

## Competing interests

Y.W., W.Z. and H-X.L. are employees at Zhuhai Trinomab Biotechnology Co., Ltd. The remaining authors declare no competing interests.
