## [Peer Review File · Nature Communications]

REVIEWER COMMENTS

Reviewer #1 (Remarks to the Author):

Kang et al. sorted B cells that express SARS-CoV-2 N and S-specific antibodies from COVID-19 patients. They picked 32 antibodies and further characterized three antibodies that can recognize SARS-CoV-2 N protein with high affinity. The authors crystallize the complex of SARS-CoV-2 N protein with one of the monoclonal antibodies and showed that changes in the epitopes and antigen's allosteric regulation. They set up the ex vivo assay to measure complement hyperactivation. The authors showed the monoclonal antibody inhibits SARS-CoV-2 N protein-induced complement hyperactivation. They performed the structural and functional analysis for the SARS-CoV-2 N-specific antibody.

They provide very interesting and important data for SARS-CoV-2 N-specific antibody response in patients. I have some comments.

1. The authors picked nCoV454 as one of the seven mAbs that bind to N-NTD, but actually it did not show N-NTD binding (in Figure 2b). Please clarify.
2. Do 18 monoclonal antibodies, which do not bind to N-NTD or N-CTD, bind to the rest of portion (AA 1-40, 175-249,365-419)?
3. Figure 1c and 1d and their figure legend are not clear. Do you need to stain two colors with the same antigens? You may also need to explain clearly in the main text (line 115-118).

Minor point:

Line 178, should "the nCoV396 Fab nCoV396Fab" be "nCoV396 Fab"?

Reviewer #2 (Remarks to the Author):

Comments:

Kang et al. in their manuscript, 'A COVID-19 antibody curbs SARS-CoV-2 nucleocapsid protein-induced complement hyperactivation', describe the isolation and characterization of mAbs from a quickly recovered SARS-CoV2 infected 30-year old patient that binds to the nucleocapsid protein of the virus. Through the crystal structure they elucidate the mode of N-protein blocking by an isolated mAb. This work is complementary to the present focus on the spike protein of this virus for the development of vaccines and diagnostic.

With the rise of escape mutants, alternate approaches in diagnostic and therapeutic development efforts need consideration. This work has implications in diagnostic and therapeutic areas and provides important data linked to quick recovery, important at this stage of the global pandemic. The data overall is suited for the readership of Nature Communications and can be accepted upon fulfilment of the requested changes. The suggestions are provided to help the authors improve the analyses of certain sections of this work.

Line 110: Please rephrase to 'To take advantage of patient ZD006' to 'To maximize analysis of patient ZD006 samples' or similar.

Line 163-164: Were the mAbs purified as their respectively identified IgG clonotypes or all as IgG1? Kindly mention briefly in the results section while pointing to Methods or other relevant section.

Section 'Complex structure of mAb with N-NTD': This is an important and interesting section of the manuscript, that could provide valuable insights to the readership with additional analysis.

- For example, authors should briefly mention here, by which method they solve the crystal structure, what was used as their homology model, or if it was done using direct methods (if so details of it), if

they have renumbered the antibody according to the Kabat nomenclature, etc.

- Authors should analyze and comment on how the constriction of the linker region by the binding mAb-nCoV396 impede viral function.
- Do other N protein targeting mAbs show similar mechanisms of action in published data for SARS2, SARS, MERS etc?
- The interactions of the variable region of the mAb should be further elaborated and compared to other published structural data.

Extended data table 4: Please mention the descriptions of the statistical terms under the table. Is I/σ really that or $\langle I \rangle / \langle \sigma \rangle$? Also, please list data collection parameters and further refinement statistics on the Statistics Table (instead of the Methods), like Ramachandran score, for proper evaluation of data quality.

Lines 210-: The authors hypothesize and biophysically characterize cross-interaction between mAb-nCoV396 and two other related coronaviruses. Is there evidence of cross-neutralization of the same viruses with this mAb?

Extended figure 2: Please explain the special characters in the consensus sequence and “TT” label on the N-NTD sequence.

Figure 3: There is considerable movement between the C-terminal region of the unbound and mAb-bound N-NTD states. The authors suggest allosteric modulation on the full-length N-protein to be a consequence of this movement. Have they (or others) experimentally observed such changes that links directly to function? How does the enlarged RNA binding pocket modulate viral function?

Extended data table 1: It would be very helpful if the number of analyzed heavy chain-light chain pairs isolated from the respective recovered patients would be tabulated on this table.

Discussion: Comprehensive analysis needs to be included comparing this present data to other publications and available data against the N-protein of SARS CoV2 and related coronaviruses like SARS or MERS. Structural, immunological and biophysical data available for relevant comparisons should be discussed.

Reviewer #3 (Remarks to the Author):

General Comments: In this manuscript, Kang et al utilize the convalescent plasma of 6 patients who recovered from SARS-CoV2 infection to profile the antibody responses in the early recovery phase of infection. They subsequently focus on a patient who had a rapid recovery (ZD006), and identify that this patient had dominant antibody responses to the SARS-CoV2 N-protein. They conduct detailed structural analyses for these antibodies, which provide novel insights into how one of these antibodies (nCoV396) binds to the RNA binding domain of the N-protein and subsequently, provide evidence (using a virus-free in vitro system) how a specific monoclonal antibody against the N protein may affect N protein-induced complement hyper activation. This manuscript advances the field in helping understand the interaction of the mAb to the N protein RNA binding domain and how it may potentially modulate disease severity, in which lies its conceptual innovation. Their in vitro systems provide technical innovation, although have inherent limitations of a virus free system. Overall, the statistical analysis is appropriate and valid. Reproducibility of some of the functional assays could be improved as detailed below.

MAJOR COMMENTS:

1. The authors use a C2 internal quenching fluorescent peptide-based analysis for ex vivo complement activation. However, this system needs greater descriptive detail. For example, is this system truly representative of MASP-2 protease mediated cleavage of C2, and if so, how [in terms of design, validation]?

2. Towards the end of the results section, the authors state "in conclusion, these results demonstrate that ...not only by facilitating Vmax of MASP-2 catalytic activity but also enhancing substrate binding specificity in the reactions". Is there an independent way to validate these findings, and how can they be certain this is the case? For example, is there a positive control for MASP-2 catalytic activity that has been utilized? Either that, or another assay for MASP-2 mediated cleavage would be important. Along those lines, is there a way to assess how the binding of nCoV396 to the N-protein modulates inflammation (including but not restricted to markers of complement activation) either in vivo or ex vivo?

3. The authors use five other serum samples from autoimmune disease donors. These samples require additional description. For example, what is 'autoimmune disease' and why were these chosen? How do they compare to healthy donors? And how do existing levels of complement activation markers and serine proteases (including MASP-2) in these specimens modulate the functional analysis?

4. The authors selected nCoV396, nCoV416 and nCoV457 for production of recombinant Fab antibodies for purposes of functional and structural characterization. It is unclear why they subsequently chose to focus on nCoV396 for purposes of testing complement hyper activation. Do they have data on the two other antibodies and do those results affect the conclusions?

MINOR COMMENTS:

1. Page 4 - the authors state "a recent retrospective observational study ofrevealed that complement disorder was associated with"....Please be more specific regarding this observation.

2. Page 4 - A recent preprint study....it is good that the authors have acknowledged this study is a preprint study. It is somewhat concerning this study was uploaded on a preprint server in March 2020 but has yet not been published, although is widely cited. To that extent, at the end of this paragraph, the authors should add a sentence about how the current study can help further advance the role of N-protein inducing complement activation. They should also subsequently address this in detail in the discussion, possibly via focusing on the RNA binding domain of the N-protein. Do the authors suggest that this domain is what is contributing to complement-mediated hyper activation? Or how do they put the findings of their current study, into perspective with the two prior studies (references 19 and 21)?

3. The authors mention that ZD004 and ZD006 had only minimal levels of antibody response to the S protein, but much higher titers to the N protein. Hence, how likely is the current schema proposed valid in the human population? The authors may want to consider incorporating references such as McAndrews et al. (<https://insight.jci.org/articles/view/142386>), among others, into their Discussion to address this issue.

4. Regarding the antibody repertoires, how did the authors ensure that there was no cross-reactivity of monoclonal Abs to other viral proteins? Additionally, among 32 mAbs that bound to N-FL, 13 Abs bound to N-NTD and one Ab bound to N-CTD, but which part of the protein did the remaining 18 Abs bind to?

Hrishikesh Kulkarni MD, MSCI

Authors' point-by-point responses to reviewers' comments

Re: A COVID-19 antibody curbs SARS-CoV-2 nucleocapsid protein-induced complement hyperactivation (Nature Communications manuscript NCOMMS-20-45166)

We are grateful to all the reviewers for their helpful comments and suggestions for improving this manuscript. Our point-by-point responses to their comments are provided below, the reviewer's critiques are in black, and **our responses are in red**. The appropriate sentences/figures/table has been edited in the revised version of the manuscript.

Reviewer #1: Kang et al. sorted B cells that express SARS-CoV-2 N and S-specific antibodies from COVID-19 patients. They picked 32 antibodies and further characterized three antibodies that can recognize SARS-CoV-2 N protein with high affinity. The authors crystalize the complex of SARS-CoV-2 N protein with one of the monoclonal antibodies and showed that changes in the epitopes and antigen's allosteric regulation. They set up the *ex vivo* assay to measure complement hyperactivation. The authors showed the monoclonal antibody inhibits SARS-CoV-2 N protein-induced complement hyperactivation. They performed the structural and functional analysis for the SARS-CoV-2 N-specific antibody. They provide very interesting and important data for SARS-CoV-2 N-specific antibody response in patients. I have some comments.

Response: We thank this reviewer for appreciating the novelty of the first structure of SARS-CoV-2 N protein bound to a human monoclonal antibody. We are also grateful that the reviewer highlighted the coupling of structural work with *ex vivo* functional assays in this study.

1. The authors picked nCoV454 as one of the seven mAbs that bind to N-NTD, but actually it did not show N-NTD binding (in Figure 2b). Please clarify.

Response: We apologize for this mistake and thank this reviewer for suggestions to correct the manuscript. NCoV454 indeed binds to full-length N protein instead of the N-NTD. We have updated the sentence involving nCoV454 at line 164 -165 in the revised manuscript.

2. Do 18 monoclonal antibodies, which do not bind to N-NTD or N-CTD, bind to the rest of portion (AA 1-

40, 175-249,365-419)?

Response: The reviewer raises an important point in the other monoclonal antibodies. However, it is tough to express the rest portions of N protein separately alone since these regions belong to disorder or flexible parts of the protein. The comprehensive studies have suggested that the compact functional domains N protein are its NTD and CTD. These two domains play several essential roles in viral RNA recognition^{1,2}, viral genomic RNA packing³, high-order structure formation of viral ribonucleoproteins (RNP)⁴, etc. Therefore, we focused our subsequent studies on the monoclonal antibodies which bind to N-NTD, N-CTD, or full-length protein in this project.

3. Figure 1c and 1d and their figure legend are not clear. Do you need to stain two colors with the same antigens? You may also need to explain clearly in the main text (line 115-118).

Response: We have corrected the legend of Figure 1c and 1d in the revised manuscript. For Figure 1c legend, we should have indicated '*CD38 and CD27 double-positive B cells' as CD38^{hi} and CD27^{hi} double-positive B cells that have been described in the Method.*' For Figure 1d, we should have modified as '*single N and S protein-specific memory B cells showing elevated fluorescence for both fluorophores were sorted into single well of 96-well plates*'. We should have described in more detail of the sorting of the antigen-specific memory B cell. We have now added the description below in the Method section of 'Sorting of single plasma cells and memory B cells by FACS' at line 508-515 as 'To minimize false positives in the sorting of antigen-specific memory B cells, streptavidin was labeled separately with Phycoerythrin-canin7 (PE-Cy7) and Brilliant Violet 421 (BV421). Labeling with each fluorophore was carried out on separate aliquots of streptavidin, which were then mixed together prior to interaction with biotinylated the S1 and N proteins used for sorting. Cells showing elevated fluorescence for both PE-Cy7- and BV421-labeled S or N protein were sorted into single well of 96-well plates.' The relevant sentences have been revised at line 117-118.

4. Minor point: Line 178, should "the nCoV396 Fab nCoV396Fab" be "nCoV396 Fab"?

Response: We apologize for our incorrect sentence and concur with the reviewer's suggestion. The sentence has been corrected in the revised version of manuscript.

Reviewer #2: Kang et al. in their manuscript, 'A COVID-19 antibody curbs SARS-CoV-2 nucleocapsid

protein-induced complement hyperactivation', describe the isolation and characterization of mAbs from a quickly recovered SARS-CoV2 infected 30-year old patient that binds to the nucleocapsid protein of the virus. Through the crystal structure they elucidate the mode of N-protein blocking by an isolated mAb. This work is complementary to the present focus on the spike protein of this virus for the development of vaccines and diagnostic.

With the rise of escape mutants, alternate approaches in diagnostic and therapeutic development efforts need consideration. This work has implications in diagnostic and therapeutic areas and provides important data linked to quick recovery, important at this stage of the global pandemic. The data overall is suited for the readership of Nature Communications and can be accepted upon fulfilment of the requested changes. The suggestions are provided to help the authors improve the analyses of certain sections of this work.

Response: We thank this reviewer for considering our contribution to be very exciting and our experimentation to be greatly sound and it is suited for Nature Communications.

1. Line 110: Please rephrase to 'To take advantage of patient ZD006' to 'To maximize analysis of patient ZD006 samples' or similar.

Response: We concur with the reviewer's suggestion and have rephrased this sentence.

2. Line 163-164: Were the mAbs purified as their respectively identified IgG clonotypes or all as IgG1? Kindly mention briefly in the results section while pointing to Methods or other relevant section.

Response: We thank the reviewer's valuable suggestion. All mAbs were produced as IgG1 antibodies regardless of their original Ig isotypes (Reference 30 in manuscript). We have added this sentence 'All of the purified antibodies were produced as IgG1 antibodies regardless their original Ig isotypes' at Line 123-124 and a slightly more detailed description in the Method section.

3. Section 'Complex structure of mAb with N-NTD': This is an important and interesting section of the manuscript, that could provide valuable insights to the readership with additional analysis.

(1) For example, authors should briefly mention here, by which method they solve the crystal structure, what was used as their homology model, or if it was done using direct methods (if so details of it), if they

have renumbered the antibody according to the Kabat nomenclature.

Response: We concur with the reviewer's suggestion and add the following sentence at Line 181-184: 'Briefly, the complex structure was determined by molecular replacement using the N-NTD structure (PDB ID: 6M3M) and monoclonal antibody omalizumab Fab (PDB ID: 6TCN) as the search models.' and Line: 187-189: 'For highlighting the complementary determining regions (CDRs), the Kabat nomenclature is aligned in the Supplementary Fig. 2 as well.' The appropriate sentence has been edited from the revised version.

a nCoV396 Variable Domain

>L

```

1 QLVLTSQSPASASLQASVSLTCTLSSGHSNYAIAWHQQQPEKGPRLMKVNSDGSHTKGD 60
61 GIPDRFSGSSSGAERYLTISLQSEDEADYYCQTWGTGIQVFGGGTKLTVLGGPKAAPSV 120
121 TLFPPSSEELQANKATLVCLISDFYPGAVTVAWKADSSPVKAGVETTPSKQSNKYAAS 180
181 SYLSLTPEQWKSHRSYSQCQVTHEGSTVEKTVAPTECS

```

>H

```

1 QVQLVESGGGVVQPGRSLRLSCAASGFTFSSYIMHWVROAPGKGLEWVAVISYDGSNEAY 60
61 ADSVKGRFTISRDNKNTLYLQMSLRAEDTGVVYCARETGDYSSSWYDSWGRGTLVTVS 120
121 SASTKGPSVFLAPSSKSTSGGTAALGCLVKDYFPEPVTVSWNSGALTSGVHTFPAVLQS 180
181 SGLYSLSSVVTVPSSSLGTQTYICNVNHKPSNTKVDKRVPEKSCDK

```

b nCoV396 Numbering
L Light Chain

Q	L	V	L	T	Q	S	P	S	A	S	A	S	L	G	A	S	V	K	L
1	2	3	4	5	6	7	8	9	11	12	13	14	15	16	17	18	19	20	21
T	C	T	L	S	S	G	H	S	N	Y	A	I	A	W	H	Q	Q	Q	P
22	23	24	25	26	27	28	29	30	30A	31	32	33	34	35	36	37	38	39	40
E	K	G	P	R	Y	L	M	K	V	N	S	D	G	S	H	T	K	G	D
41	42	43	44	45	46	47	48	49	50	51	52	53	54	54A	54B	54C	54D	55	56
G	I	P	D	R	F	S	G	S	S	S	G	A	E	R	Y	L	T	I	S
57	58	59	60	61	62	63	64	65	66	67	68	69	70	71	72	73	74	75	76
S	L	Q	S	E	D	E	A	D	Y	Y	C	Q	T	W	G	T	G	I	Q
77	78	79	80	81	82	83	84	85	86	87	88	89	90	91	92	93	94	95	96
V	F	G	G	G	T	K	L	T	V	L	G	Q	P	K	A				
97	98	99	100	101	102	103	104	105	106	106A	107	108	109	110	111				

H Heavy Chain

Q	V	Q	L	V	E	S	G	G	G	V	V	Q	P	G	R	S	L	R	L
1	2	3	4	5	6	7	8	9	10	11	12	13	14	15	16	17	18	19	20
S	C	A	A	S	G	F	T	F	S	S	Y	I	M	H	W	V	R	Q	A
21	22	23	24	25	26	27	28	29	30	31	32	33	34	35	36	37	38	39	40
P	G	K	G	L	E	W	V	A	V	I	S	Y	D	G	S	N	E	A	Y
41	42	43	44	45	46	47	48	49	50	51	52	52A	53	54	55	56	57	58	59
A	D	S	V	K	G	R	F	T	I	S	R	D	N	S	K	N	T	L	Y
60	61	62	63	64	65	66	67	68	69	70	71	72	73	74	75	76	77	78	79
L	Q	M	S	S	L	R	A	E	D	T	G	V	Y	Y	C	A	R	E	T
80	81	82	82A	82B	82C	83	84	85	86	87	88	89	90	91	92	93	94	95	96
G	D	Y	S	S	S	W	Y	D	S	W	G	R	G	T	L	V	T	V	S
97	98	99	100	100A	100B	100C	100D	101	102	103	104	105	106	107	108	109	110	111	112
S	A																		
113	114																		

Supplementary Figure 2 | CDRs delimitation and Kabat nomenclature of the mAb nCoV396.
(a) CDRs are highlighted in colors (CDR1(red), CDR2(orange), CDR3(green)). The gray letters indicate the non-variable-domain region. The underlined black letters indicate variable domain of heavy chain while the other black letters indicate variable domain of light chain. **(b)** The CDRs are highlighted in pink (CDR1, CDR2, CDR3). The green letters indicate heavy chain region while the blue letters indicate light chain region.

(2) Authors should analyze and comment on how the constriction of the linker region by the binding mAb-nCoV396 impede viral function.

Response: We are grateful that this reviewer appreciates novel aspects of the overall structural stabilization in the context of the linker region's constriction. From the reported NTD structures of SARS-CoV (45-181aa; PDB: 1SSK), MERS (1-164aa; PDB:4UD1), and MHV (60-197aa PDB:3HD4) N protein, we found that they all contain the regions which are corresponding to 162-170 residues of SARS-CoV-2. These results indicate that 162-170 residues are indispensable for the structure and function of N protein. We also found that 162-170 is the recognized region for the antibodies binding to NTD or full-length of N protein through N-derived epitope peptide ELISA analysis (**Supplementary Table 1**). These data indicated that 162-170 of SARS-CoV-2 N protein is a vital epitope. The appropriate sentence (line 280-287) has been edited from the revised version: *'To validate that any other N-specific mAbs have similar functions as nCoV396, we next sought to identify the linear epitope of the mAbs to N protein. Briefly, 68 overlapping 18-mer peptides derived from the full-length N-protein have been synthesized and used for epitope screening. The binding capacity of the mAb candidate to the epitopes was analyzed by enzyme-linked immunosorbent assay (ELISA). Consistently with nCoV396, several mAbs (nCoV454, nCoV457) display positivity against the peptide epitopes of N protein located between 157-180 residues (Supplementary Table 1).'*

Supplementary table 1: Epitope Mapping of SARS-CoV-2 N protein antibodies by Elisa

Peptide Number	Peptide sequence (N→C)	nCOV 0396			nCOV 0454			nCOV 0457			TT0170		
NP1(1-18aa)	MSDNGPQNQRNAPRITFGGSK (Biotin)	0.021	0.016	0.013	0.022	0.022	0.029	0.049	0.038	0.035	0.011	0.011	0.015
NP2(7-24aa)	QNQRNAPRITFGGSDSTGSK (Biotin)	0.014	0.009	0.005	0.016	0.016	0.027	0.039	0.029	0.026	0.004	0.004	0.012
NP3(13-30aa)	PRITFGGSDSTGSNQNNGSK (Biotin)	0.016	0.009	0.007	0.018	0.022	0.031	0.041	0.032	0.028	0.006	0.008	0.015
NP4(19-36aa)	GFSDSTGSNQNNGERSGARGSK (Biotin)	0.017	0.011	0.007	0.022	0.023	0.032	0.041	0.032	0.028	0.008	0.011	0.017
NP5(25-42aa)	GSNQNNGERSGARGSKQRRPFGSK (Biotin)	0.019	0.013	0.008	0.023	0.022	0.029	0.037	0.03	0.028	0.009	0.012	0.015
NP6(31-48aa)	ERSGARGSKQRRPGLPNNNGSK (Biotin)	0.018	0.012	0.005	0.022	0.023	0.037	0.041	0.03	0.025	0.008	0.01	0.014
NP7(37-54aa)	SKQRRPQLPNNNTASWFTGSK (Biotin)	0.028	0.015	0.017	0.03	0.027	0.032	0.042	0.031	0.03	0.012	0.01	0.017
NP8(43-60aa)	QGLFNNTASWFTALTQHGGSK (Biotin)	0.021	0.013	0.015	0.028	0.03	0.037	0.026	0.024	0.022	0.008	0.007	0.011
NP9(49-66aa)	TASWFTALTQHGKEDLKFSGSK (Biotin)	0.023	0.017	0.016	0.027	0.039	0.038	0.034	0.026	0.023	0.007	0.012	0.014
NP10(55-72aa)	ALTHGKEDLKFPRGQVGVGSK (Biotin)	0.027	0.016	0.017	0.028	0.036	0.047	0.039	0.031	0.024	0.007	0.012	0.015
NP11(61-78aa)	KEDLKFPRGQVGPINTNSGSK (Biotin)	0.025	0.016	0.016	0.031	0.033	0.045	0.032	0.025	0.022	0.007	0.008	0.015
NP12(67-84aa)	PRGQVGPINTNSPDDQIGSK (Biotin)	0.024	0.016	0.015	0.031	0.036	0.046	0.041	0.026	0.023	0.007	0.006	0.014
NP13(73-90aa)	PINTNSPDDQIGYYRAGSK (Biotin)	0.029	0.025	0.025	0.03	0.033	0.033	0.066	0.06	0.056	0.036	0.03	0.028
NP14(79-96aa)	SPDDQIGYYRRARRIRGSK (Biotin)	0.025	0.025	0.026	0.02	0.024	0.027	0.061	0.044	0.038	0.023	0.026	0.032
NP15(85-102aa)	GYRRARRIRIRGGDKMKGSK (Biotin)	0.03	0.028	0.023	0.023	0.025	0.028	0.067	0.051	0.047	0.029	0.035	0.037
NP16(91-108aa)	TRRIRGGDKMKDLSFRWGSK (Biotin)	0.033	0.03	0.027	0.025	0.026	0.029	0.059	0.053	0.056	0.034	0.035	0.038
NP17(97-114aa)	GDGKMKDLSFRWFYLLGGSK (Biotin)	0.026	0.034	0.031	0.024	0.028	0.031	0.065	0.056	0.061	0.04	0.034	0.044
NP18(103-120aa)	DLSFRWFYLLGTGPEAGGSK (Biotin)	0.031	0.027	0.03	0.026	0.027	0.03	0.062	0.056	0.063	0.031	0.032	0.041
NP19(109-126aa)	YFYYLGTGPEAGLPGANGSK (Biotin)	0.035	0.024	0.024	0.032	0.026	0.033	0.044	0.037	0.044	0.016	0.015	0.042
NP20(115-132aa)	TGPEAGLPGANKDGIWGSK (Biotin)	0.021	0.015	0.012	0.028	0.025	0.036	0.045	0.031	0.026	0.013	0.008	0.03
NP21(121-138aa)	LPYGANKDGIWVATEGAGSK (Biotin)	0.027	0.018	0.014	0.029	0.023	0.036	0.04	0.03	0.031	0.012	0.009	0.027
NP22(127-144aa)	KDGIWVATEGALNTPKDGSK (Biotin)	0.026	0.018	0.012	0.031	0.029	0.046	0.044	0.03	0.032	0.013	0.013	0.032
NP23(133-150aa)	VATEGALNTPKDHIGTRNGSK (Biotin)	0.022	0.016	0.011	0.033	0.031	0.041	0.044	0.032	0.036	0.013	0.012	0.034
NP24(139-156aa)	LNTPKDHIGTRNPANNAAGSK (Biotin)	0.021	0.011	0.01	0.031	0.033	0.049	0.047	0.039	0.033	0.013	0.014	0.031
NP25(145-162aa)	HIGTRNPANNAIVLQLPGSK (Biotin)	0.031	0.027	0.026	0.052	0.044	0.054	0.036	0.034	0.033	0.011	0.017	0.023
NP26(151-168aa)	PANNAIVLQLPQGTTLPGSK (Biotin)	0.02	0.023	0.022	0.041	0.04	0.054	0.041	0.031	0.028	0.007	0.01	0.015
NP27(157-174aa)	IVLQLPQGTTLPGFYAEGSK (Biotin)	2.439	2.02	2.34	1.686	1.79	2.03	2.193	2.142	1.648	0.007	0.011	0.018
NP28(163-180aa)	QGTTLPGFYAEGSRGGSGSK (Biotin)	2.215	2.007	1.83	1.86	1.844	2.216	2.047	1.795	1.601	0.009	0.014	0.02
NP29(169-186aa)	KGFYAEGSRGGSSASSRSGSK (Biotin)	0.03	0.023	0.028	0.043	0.042	0.059	0.034	0.031	0.023	0.011	0.018	0.02
NP30(175-192aa)	GSRGGSSASSRSSSRNRGSK (Biotin)	0.028	0.021	0.023	0.04	0.038	0.069	0.037	0.029	0.022	0.008	0.012	0.018
NP31(181-198aa)	QASSRSSSRNRSSRNSTGSK (Biotin)	0.018	0.022	0.027	0.034	0.031	0.042	0.045	0.045	0.046	0.037	0.032	0.041
NP32(187-204aa)	SSRNRSSRNSTPGSSRGGSK (Biotin)	0.018	0.03	0.024	0.023	0.026	0.042	0.047	0.04	0.042	0.024	0.023	0.035
NP33(193-210aa)	SSRNSTPGSSRGTSPARMGSK (Biotin)	0.021	0.028	0.023	0.024	0.027	0.043	0.051	0.048	0.049	0.028	0.024	0.036
NP34(199-216aa)	PGSSRGTSPARMAGNGDGSK (Biotin)	0.027	0.023	0.024	0.026	0.027	0.043	0.054	0.052	0.058	0.033	0.029	0.047
NP35(205-222aa)	TSPARMAGNGDAAALALLGSK (Biotin)	0.026	0.03	0.029	0.029	0.03	0.044	0.054	0.048	0.053	0.035	0.029	0.045
NP36(211-228aa)	ACNGGDAALALLLDRLNGSK (Biotin)	0.02	0.031	0.031	0.025	0.032	0.039	0.051	0.048	0.053	0.041	0.03	0.045
NP37(217-234aa)	AALALLLDRLNQLSKMGSK (Biotin)	0.015	0.013	0.015	0.032	0.03	0.043	0.045	0.034	0.029	0.01	0.012	0.015
NP38(223-240aa)	LLDRLNQLSKMSGKQGGSK (Biotin)	0.012	0.012	0.01	0.026	0.025	0.039	0.036	0.031	0.024	0.006	0.005	0.012
NP39(229-246aa)	QLSKMSGKQGGQQTGSK (Biotin)	0.019	0.013	0.012	0.026	0.029	0.035	0.038	0.032	0.028	0.011	0.011	0.016
NP40(235-252aa)	SGKQGGQQTGVTKSAAGSK (Biotin)	0.019	0.014	0.013	0.032	0.034	0.041	0.041	0.034	0.028	0.013	0.013	0.017
NP41(241-258aa)	QQGQTVTKKSAEASKKPGSK (Biotin)	0.024	0.016	0.013	0.033	0.035	0.044	0.041	0.036	0.028	0.012	0.011	0.015
NP42(247-264aa)	TKKSAEASKKPRKRTAGSK (Biotin)	0.017	0.013	0.012	0.028	0.029	0.033	0.045	0.036	0.026	0.01	0.01	0.013
NP43(253-270aa)	EASKKPRKRTATKAYNVGSK (Biotin)	0.02	0.016	0.013	0.038	0.038	0.041	0.056	0.044	0.048	0.023	0.022	0.029
NP44(259-276aa)	RQKRTATKAYNVTQAFGRGSK (Biotin)	0.019	0.01	0.007	0.03	0.023	0.044	0.062	0.036	0.032	0.012	0.012	0.025
NP45(265-282aa)	TEAYNVTQAFGRGPEQTGSK (Biotin)	0.022	0.012	0.008	0.035	0.033	0.041	0.048	0.039	0.034	0.01	0.011	0.022
NP46(271-288aa)	TQAFGRGPEQTQGNFGDGSK (Biotin)	0.022	0.009	0.01	0.035	0.036	0.035	0.052	0.039	0.034	0.011	0.011	0.022

NP47(277-294aa)	RGPEQTQGNFGDQELIRQGSX(Biotin)	0.018	0.013	0.01	0.044	0.038	0.04	0.05	0.039	0.035	0.013	0.012	0.022
NP48(283-300aa)	QGNFGDQELIRQGTDYKHGSK(Biotin)	0.016	0.012	0.009	0.041	0.035	0.039	0.058	0.04	0.037	0.015	0.014	0.024
NP49(289-306aa)	QELIRQGTDYKHWPQIAQGSX(Biotin)	0.016	0.012	0.017	0.023	0.028	0.037	0.061	0.048	0.041	0.021	0.026	0.02
NP50(295-32aa)	GTDYKHWPQIAQFAPSASGSX(Biotin)	0.015	0.009	0.012	0.023	0.024	0.039	0.062	0.038	0.033	0.021	0.018	0.024
NP51(301-318aa)	WPQIAQFAPSASAFPMSGX(Biotin)	0.015	0.014	0.026	0.024	0.025	0.039	0.066	0.043	0.061	0.025	0.027	0.031
NP52(301-324aa)	FAPSASAFPMSGRIGMEVGSX(Biotin)	0.015	0.02	0.022	0.027	0.029	0.041	0.061	0.045	0.054	0.031	0.027	0.024
NP53(313-330aa)	AFPMSGRIGMEVTFPGTWGSX(Biotin)	0.017	0.016	0.023	0.025	0.027	0.031	0.06	0.04	0.051	0.029	0.023	0.027
NP54(319-336aa)	RIGMEVTFPGTWLTYTAAGSX(Biotin)	0.016	0.014	0.017	0.024	0.026	0.036	0.048	0.049	0.044	0.03	0.024	0.036
NP55(325-343aa)	TFPGTWLTYTAAIKLDDKGSX(Biotin)	0.037	0.022	0.023	0.033	0.028	0.033	0.042	0.04	0.04	0.023	0.018	0.023
NP56(331-348aa)	LTYTAAIKLDDKDPNFKDGSX(Biotin)	0.03	0.012	0.011	0.028	0.023	0.036	0.038	0.036	0.032	0.019	0.016	0.03
NP57(337-354aa)	IKLDDKDPNFKDQVILLNGSX(Biotin)	0.033	0.017	0.01	0.028	0.028	0.036	0.045	0.042	0.035	0.019	0.018	0.03
NP58(343-360aa)	DPNFKDQVILLNKHIDAYGSX(Biotin)	0.031	0.017	0.012	0.032	0.029	0.039	0.047	0.043	0.036	0.021	0.019	0.032
NP59(349-366aa)	QVILLNKHIDAYKTFPPTGSX(Biotin)	0.03	0.013	0.009	0.034	0.031	0.04	0.043	0.039	0.04	0.021	0.02	0.035
NP60(355-372aa)	KHIDAYKTFPPTFPKDKGSX(Biotin)	0.032	0.01	0.006	0.033	0.03	0.04	0.04	0.04	0.038	0.021	0.02	0.036
NP61(361-378aa)	KTFPPTFPKDKKKKADGGSX(Biotin)	0.034	0.021	0.022	0.041	0.038	0.044	0.038	0.032	0.034	0.011	0.016	0.023
NP62(367-384aa)	EPKDKKKKADDTQALPQGSX(Biotin)	0.024	0.015	0.013	0.034	0.038	0.041	0.04	0.027	0.027	0.008	0.007	0.018
NP63(373-390aa)	KKKADDTQALPQRQKQGSX(Biotin)	0.02	0.015	0.014	0.039	0.04	0.053	0.035	0.023	0.025	0.004	0.007	0.021
NP64(379-396aa)	TQALPQRQKQQTIVLLEGSX(Biotin)	0.021	0.015	0.013	0.041	0.042	0.053	0.037	0.024	0.022	0.008	0.006	0.018
NP65(385-402aa)	RQKQQTIVLLEPAADLDDGSX(Biotin)	0.026	0.018	0.014	0.036	0.037	0.047	0.036	0.023	0.021	0.008	0.008	0.018
NP66(391-408aa)	TVLLEPAADLDDFSKQLQGSX(Biotin)	0.02	0.015	0.011	0.038	0.043	0.06	0.045	0.026	0.025	0.007	0.009	0.021
NP67(397-414aa)	AADLDDFSKQLQSSMSAGSX(Biotin)	0.02	0.014	0.009	0.032	0.026	0.04	0.046	0.031	0.029	0.01	0.01	0.012
NP68(403-419aa)	FSKQLQSSMSADSTQAAGSX(Biotin)	0.016	0.01	0.007	0.027	0.028	0.031	0.039	0.03	0.029	0.007	0.007	0.011

Reactivity of the antibodies against synthetic peptides measured by ELISA.
The highlighted part indicates that the antibody binds to this peptide.

Furthermore, we also verified that other two antibodies (nCoV454 and nCoV457) with the same binding site as nCoV396 could inhibit the complement hyperactivation caused by SARS-CoV-2 N protein. The experimental results are shown in **Supplementary Fig. 3a, b**. Therefore, we believe that the region of 162-170 is very important for the virus function.

Supplementary Figure 3 | Other antibodies (nCoV454 and nCoV457) compromise SARS-CoV-2 and SARS-CoV N-induced complement hyperactivation (a) The Michealis-Menten curve of *ex vivo* SARS-CoV-2 N protein-induced excessive cleavage of C2 in the serum. The v_0 is indicated on the Y-axis, the concentration of substrate is indicated on the X-axis. mAb nCoV396 (dark blue), nCoV457 (cyan), nCoV454 (black), Negative control with other protein (ENL) expressed in *E. coli* (orange), and Blank control (blue) are presenting. **(b)** the corresponding kinetics parameters of **(a)** are presented. **(c)** The Michealis-Menten curve of *ex vivo* SARS-CoV N protein-induced excessive cleavage of C2 in the serum, with similar representation as **(a)**. **(d)** The corresponding kinetics parameters of **(c)** are presented.

(3) Do other N protein targeting mAbs show similar mechanisms of action in published data for SARS2, SARS, MERS, etc.?

Response: To our knowledge, the ability of N-specific mAbs in COVID-19 against hyperactivation of the complement system is firstly reported by our group. Our supplementary data supports that not only the nCoV396 is capable of inhibition of N-induced complement hyperactivation, but also other two mAbs (nCoV454 and nCoV457) can work on it (Please see the response for Reviewer #2 point 3-(2)). To further validate whether these mAbs work against other highly pathogenic relative N proteins in the ex vivo complement system, we next perform the ex vivo assays in the presence of SARS-CoV N-protein. As shown in **Supplementary Fig. 3c, d**, the SARS-CoV-2 N-directed mAbs (nCoV396, nCoV454, and nCoV457) display potent inhibition to SARS-CoV N protein-induced MASP-2 hyperactivation with decreased V_{max} in the assays (line 293-299). Therefore, our data support N protein's similar mechanisms in SARS-CoV, and the mAbs isolated in this work function as effective inhibitors for N protein-induced complement hyperactivation.

(4) The interactions of the variable region of the mAb should be further elaborated and compared to other published structural data.

Response: To our knowledge, the structural study reported here is the first crystal structure of N-specific antibody in complex with N protein. Therefore, we can't compare it to other structural data. In the Extend Data Figure 1, we have shown the N protein and nCoV396 interaction interface. To explain more detailed interactions of the mAb with N-NTD, we have added the following sentences into the revised manuscript (Line 205-207). 'Briefly, the residues G27, Y31, A32, W95, G98, I99 of variable region V_L bound to 159-163 of N protein, whereas the residues I33, V50, N57, A59, E99, T100, D102, Y103, S105, S106 of variable region V_H bound to 165-172 of N protein.'

To evaluate the potential epitope of other mAbs, we perform the epitope screening assays with continually N-derived peptides (Please see the response for Reviewer #2 point 3-(2)). As shown in **Supplementary Table 1**, the most potent epitope is the sequence of 157-180 amino acids, consistently with our reported data in the manuscript. The appropriate sentence has been edited from the revised version.

4. Extended data table 4: Please mention the descriptions of the statistical terms under the table. Is I/s really that or /<s>? Also, please list data collection parameters and further refinement statistics on the

Statistics Table (instead of the Methods), like Ramachandran score, for proper evaluation of data quality.

Response: I/s means I/<s>. We concur with the reviewer's suggestion and have corrected it using this table in the revised version of manuscript.

Extended data table 4. Data collection and refinement statistics

The Complex of mAb-396 with SARS-CoV-2 N-NTD (47-175)*	
Data collection	SSRF BL-18U (PDB:7CR5)
Space group	P 2 ₁ 2 ₁ 2
Cell dimensions	
a , b , c (Å)	154.07, 52.60, 85.30
α , β , γ (°)	90, 90, 90
Resolution (Å)	50 - 2.1 (2.14 - 2.1) **
$R_{\text{merge}}^{\#}$	0.22 (1.33)
I / $\sigma(I)$	16.07 (1.64)
Completeness (%)	99.5 (93.7)
Redundancy	12.1 (8.4)
Refinement	
Resolution (Å)	28.43 - 2.1 (2.16 - 2.1)
No. reflections	41691 (3561)
$R_{\text{work}}/R_{\text{free}}$ (%) ^{##}	0.19 / 0.22
No. atoms	4434
Protein	4178
Ligand/ion	1
Water	255
B -factors (Å ²)	35.39
Protein	35.19
Ligand/ion	73.44
Water	38.54
R.m.s. deviations	
Bond lengths (Å)	0.007
Bond angles (°)	0.87
Ramachandran plot (%)	
Favored	97.1
Allowed	2.9
Disallowed	0.0

*This dataset is collected with one crystal.

**Values in parentheses are for the highest-resolution shell.

[#] $R_{\text{merge}} = \sum_{\text{hkl}} \sum_i |I_i(\text{hkl}) - \langle I(\text{hkl}) \rangle| / \sum_{\text{hkl}} \sum_i I_i(\text{hkl})$, where $I_i(\text{hkl})$ is the intensity measured for the i th reflection and $\langle I(\text{hkl}) \rangle$ is the average intensity of all reflections with indices hkl.

^{##} $R_{\text{work}} = \sum_{\text{hkl}} ||F_{\text{obs}}(\text{hkl}) - |F_{\text{calc}}(\text{hkl})|| / \sum_{\text{hkl}} |F_{\text{obs}}(\text{hkl})|$. R_{free} is calculated in an identical manner using 10% of randomly selected reflections that were not included in the refinement.

5. Lines 210-: The authors hypothesize and biophysically characterize cross-interaction between mAb-nCoV396 and two other related coronaviruses. Is there evidence of cross-neutralization of the same viruses with this mAb?

Response: We found that mAb-nCoV396 has a high affinity with SARS-CoV N protein and MERS-CoV N protein by SPR (**Extended Data Figure 2b**). Besides, we repeated the *ex vivo* assays with SARS. We found that SARS-CoV N protein can also induce complement hyperactivation, and mAb-nCoV396 can inhibit this induction (Please see the response for Reviewer #2 point 3-(2)).

6. Extended figure 2: Please explain the special characters in the consensus sequence and "TT" label on the N-NTD sequence.

Response: We feel sorry for not clearly describing the meaning of the symbols in alignment results and should have added the following sentence in **Extended Data Figure 2** legend: 'The η symbol refers to a 3_{10} -helix. β -strands are rendered as arrows, strict β -turns as TT letters⁵.'

7. Figure 3: There is considerable movement between the C-terminal region of the unbound and mAb-bound N-NTD states. The authors suggest allosteric modulation on the full-length N-protein to be a consequence of this movement. Have they (or others) experimentally observed such changes that links directly to function? How does the enlarged RNA binding pocket modulate viral function?

Response: We concur with the reviewer's suggestion and perform the RNA electrophoretic mobility shift assays (EMSA) for evaluating the potential functional effects upon the N protein binding RNA. As shown in **Supplementary Fig.5**, nCoV396 can significantly reduce the ability of N protein binding RNA. Shing-Yen Lin et al. found that the compound PJ34 can significantly decrease HCoV OC43 N protein's RNA-binding affinity and subsequently decrease in viral replication². In the study of other viruses, nucleotide analogs were shown to inhibit influenza A virus replication by preventing RNP formation during viral particle production⁶. These data indicate that it is a validated method of antiviral treatment through interference with the N protein's RNA-binding activity.

Supplementary Figure 5 | Electrophoretic mobility shift assays of SARS-CoV-2 N protein with viral TRS RNA. Mobility shift of biotinylated TRS RNA bound to SARS-CoV-2 N protein (**a**) and SARS-CoV-2 N protein mixed with mAb nCoV396 (Molar concentration ratio=1:1) (**b**). The protein concentration was increased by a factor of 2, starting from lane 1 (7.8125 nM) to lane 13 (32 μM). Lane C, negative control. Lane 14, the highest concentration of protein without biotinylated TRS RNA. (**c**) and (**d**) Fitting of the kinetic dissociation values of SAR-CoV-2 (**a**) and SARS-CoV-2 N protein mixed with mAb nCoV396 (**b**) based on the EMSA results.

8. Extended data table 1: It would be very helpful if the number of analyzed heavy chain-light chain pairs isolated from the respective recovered patients would be tabulated on this table.

Response: All these 32 antibodies were isolated from the same subject CoV006. The information of the antibodies is summarized in Extended Data Table 2. To further analyze the germline gene preference of the N-directed antibodies, we list the distribution of IGHV and IGκ/λV gene usages of N-specific mAbs, as shown in **Supplementary Fig. 1**. we have added the following sentences into the revised manuscript (Line 135-137) '*Various germline genes are used in N-induced antibodies from CoV006, and the germline gene usage shows strong preferences for IGHV3-30, IGκ/λV4-69, respectively*'.

Supplementary Figure 1 | The distribution of IGHV gene and IGκ/λV gene usage of SARS-CoV-2 N-reactive antibodies. A total of 32 SARS-CoV-2 N-reactive antibodies from ZD006 is analyzed. **(a)** Number of IGHV gene from 32 antibodies in this study is shown on the Y-axis, IGHV gene is indicated on the X-axis. **(b)** Number of IGκ/λV gene from 32 antibodies in this study is shown on the Y-axis, IGκ/λV gene is indicated on the X-axis.

9. Discussion: Comprehensive analysis needs to be included comparing this present data to other publications and available data against the N-protein of SARS CoV2 and related coronaviruses like SARS or MERS. Structural, immunological and biophysical data available for relevant comparisons should be discussed.

Response: We concur with the reviewer's suggestion. As mentioned in response to critique Reviewer #2 point 3-(4), the structure shown in this study is the first complex structure of N-specific with N protein. Therefore, we only compare the available immunological and biophysical data in the Discussion part of the revised version.

Reviewer #3 (Remarks to the Author):

General Comments: In this manuscript, Kang et al. utilize the convalescent plasma of 6 patients who recovered from SARS-CoV2 infection to profile the antibody responses in the early recovery phase of infection. They subsequently focus on a patient who had a rapid recovery (ZD006) and identify that this patient had dominant antibody responses to the SARS-CoV2 N-protein. They conduct detailed structural analyses for these antibodies, which provide novel insights into how one of these antibodies (nCoV396) binds to the RNA binding domain of the N-protein and subsequently, provide evidence (using a virus-free

in vitro system) how a specific monoclonal antibody against the N protein may affect N protein-induced complement hyper activation. This manuscript advances the field in helping understand the interaction of the mAb to the N protein RNA binding domain and how it may potentially modulate disease severity, in which lies its conceptual innovation. Their in vitro systems provide technical innovation, although have inherent limitations of a virus free system. Overall, the statistical analysis is appropriate and valid. Reproducibility of some of the functional assays could be improved as detailed below.

Response: We thank this reviewer for appreciating the novelty of our report about the monoclonal antibody (nCoV396) against the N protein can curb N protein-induced complement hyperactivation. We are very grateful to the reviewer for providing valuable comments on our article, and we have made a serious response to these comments.

MAJOR COMMENTS:

1. The authors use a C2 internal quenching fluorescent peptide-based analysis for ex vivo complement activation. However, this system needs greater descriptive detail. For example, is this system truly representative of MASP-2 protease mediated cleavage of C2, and if so, how [in terms of design, validation]?

Response: The mannan-binding lectin (MBL)-associated serine proteases (MASPs) circulate in serum complexed with mannan-binding lectin, a complement system's recognition molecule. MASP-2 cleaves the complement components C4 and C2 to form the C3 convertase C4b2a. The fluorescent peptide-based assay was developed using a C2-derived peptide sequence (SLGRKIQI) conjugated to Dnp fluorescent group. The synthetic fluorescent peptides were quenched by their N-terminal 2Abz group. Once cleavage occurs, which is mediated explicitly by MASP-2, the fluorescent Dnp group is released. This release can be monitored over a time course by a spectrofluorometer using an excitation wavelength of 320 nm and an emission wavelength of 420 nm. To avoid the artificial factors in serum, we next perform an *in vitro* fluorescent peptide-based assay without serum conditions. Only recombinant MASP-2 protein and C2 internal quenching fluorescent peptide mixed in a reaction buffer for *in vitro* analyzing systems. In agreement with *ex vivo* system results, our newly developed *in vitro* assay demonstrates that SARS-CoV-2 N protein induces hyperactivation of MASP-2 activity. In contrast, the addition of mAb nCoV396 inhibits the N-induced MASP-2 hyperactivation (**Supplementary Fig. 4**). Taken together, our works reveal that these systems truly representative of MASP-2 protease mediated cleavage of C2.

Supplementary Figure 4 | The effect of N protein on the enzyme efficiency of recombinant MASP-2 (287-686aa) fragments based on the fluorescent substrate peptides. (a) SDS-PAGE analysis of purified recombinant MASP-2 (287-686aa) fragments. Marker (left lane), SUMO-tag MASP-2 protein (middle lane, ranged 287-686 residues), and digested product of SUMO-tag MASP-2 protein with ULP1 protease are shown. **(b)** The Michealis-Menten curve of SARS-CoV-2 N protein-induced excessive cleavage of C2 in the presence of recombinant MASP-2 *in vitro*. The v_0 is indicated on the Y-axis, the concentration of substrate is indicated on the X-axis. The reaction system without SARS-CoV-2 N protein (dark blue), or in the presence of 4 μM SARS-CoV-2 N protein (purple), 8 μM SARS-CoV-2 N protein (magenta), 16 μM SARS-CoV-2 N protein (pink) and Negative control with other protein (ENL) expressed in *E. coli* (orange) are presenting. **(c)** The corresponding kinetics parameters of **(b)** are presented. **(d)** Fluorescence intensity of varying mAb nCoV396 concentrations added to the *in vitro* enzymatic solution. The concentrations of each component are the final concentrations seen in the list. The 320 nM MASP-2 mixture with 16 μM SARS-CoV-2 N protein (purple) and the addition of the gradient concentration nCoV396 (dark blue) or IgG (orange) as negative control; MASP-2 mixture without SARS-CoV-2 N protein as blank control. The reaction mixture was then analyzed by detecting fluorescence signal by 100 μM C2 substrate peptides and the v_0 is indicated on the Y-axis.

2. Towards the end of the results section, the authors state "in conclusion, these results demonstrate that ...not only by facilitating V_{max} of MASP-2 catalytic activity but also enhancing substrate binding specificity in the reactions". Is there an independent way to validate these findings, and how can they be certain this is the case? For example, is there a positive control for MASP-2 catalytic activity that has been utilized? Either that, or another assay for MASP-2 mediated cleavage would be important. Along those lines, is there a way to assess how the binding of nCoV396 to the N-protein modulates inflammation (including but not restricted to markers of complement activation) either in vivo or ex vivo?

Response: As described in response to Reviewer #3 critique point 1, we purified the catalytic domain of MASP2 *in vitro* and added SARS2-CoV-2 N protein with increasing concentrations. We found that the enzyme efficiency of MASP-2 increased as the concentration of N protein increased (**Supplementary Fig. 4**) please see the response for Reviewer #3 point 1). We further analyzed the variation of V_{max} , K_m , and K_{cat} , which can represent the characteristics of MASP-2 enzyme activity. Consistently with our *ex vivo* assays, the V_{max} of the in vitro reactions remarkably elevate up, with identical increased V_{max}/K_m values as well. These independent kinetic analyses support the induced MASP-2 activations by N protein. Since our *ex vivo* system has eliminated immune cells in samples, we could not continue to analyze the inflammation in our system or the antibody-mediated inflammation modulation. Supporting our results, Gao T. et al. reports other highly pathogenic coronavirus SARS-CoV N-specific antibodies to reduce the death rate and lung tissue inflammation in the LPS-mice model with adenovirus-expressing SARS-CoV N (Gao T. et al. medRxiv preprint, Figure 3A). Furthermore, the recent study of COVID-19 patients treated with Narsoplimab, a MASP-2 specific monoclonal antibody, suggested that inhibition of MASP-2 was associated with a rapid and sustained reduction of circulating endothelial cell counts and concurrent reduction of serum IL-6, IL-8, C-reactive protein, and lactate dehydrogenase ⁷. Combining with our results, SARS-CoV-2 N protein-induced MASP-2 hyperactivation plays a pivotal role in COVID-19 related inflammation.

(Gao T. et al. Preprint at <https://www.medrxiv.org/content/10.1101/2020.03.29.20041962v3>, Figure 3A)

3. The authors use five other serum samples from autoimmune disease donors. These samples require additional description. For example, what is 'autoimmune disease' and why were these chosen? How do they compare to healthy donors? And how do existing levels of complement activation markers and serine proteases (including MASP-2) in these specimens modulate the functional analysis?

Response: In extended table 5, we listed the necessary information of the enrolled patients, including gender, age, diagnosis, and the values of C3 and C4 at the time of sampling. Since we want to simulate the state of immune hyper-activation, our selection criteria are patients with abnormal C3 or C4 values. To avoid the inaccurate description, we have now edited the sentence at Line 277 as 'five other serum samples with abnormal serologic C3 or C4 values'. Additionally, we compared samples described above with healthy donor samples with normal serologic parameters in C2 internal quenching fluorescent peptide-based *ex vivo* analysis. As shown in **Supplementary Fig. 6**, the serum MASP-2 activity to C2 peptide of the healthy donor is much weaker than those from abnormal serologic samples. Principally, the C3 or C4 serologic tests are used to determine if the complement system is abnormal. Increased levels for the test result may indicate activation of the complement cascade, in which the protease such as MASP-2 is ready to work on their substrate upon regulations. To reach a maximal readout of the MASP-2 activity, we select these specimens to analyze the serine proteases. Additionally, we also perform the *in vitro* assays to support our conclusions, as shown in response to point1 and point 2 critiques of Reviewer #3.

a

b

	Health-113	Health-113+SARS-CoV-2 N	patient-81	patient-81+SARS-CoV-2 N	Negative Ctrl
Vmax (RU s ⁻¹)	-0.0699	0.1944	1.189	2.15	-0.04826
95% CI	/	0.1794-0.2114	0.8049-2.380	1.720-2.905	/
Km (µM)	92.95	13.8	327.9	117	69.69
Vmax/Km (RU s ⁻¹ /µM)	-0.0008	0.0141	0.0036	0.0184	-0.0007

Supplementary Figure 6 | The *ex vivo* SARS-CoV-2 N-induced complement hyperactivation assays in different donor. (a) The Michaelis-Menten curve of the *ex vivo* SARS-CoV-2 N protein-induced excessive cleavage of C2 in the serum of patient-81 and health-113. The v_0 is indicated on the Y-axis, the concentration of substrate is indicated on the X-axis. Health donor serum (purple), health donor serum with addition of SARS-CoV-2 N protein (dark blue), patient donor serum (green), patient donor serum with addition of SARS-CoV-2 N protein as positive control (brown) and health donor serum with addition of negative control with other protein (ENL) expressed in *E. coli* (orange) are presenting. (b) the corresponding kinetics parameters of (a) are presented.

4. The authors selected nCoV396, nCoV416 and nCoV457 for production of recombinant Fab antibodies for purposes of functional and structural characterization. It is unclear why they subsequently chose to focus on nCoV396 for purposes of testing complement hyper activation. Do they have data on the two other antibodies and do those results affect the conclusions?

Response: Since we wanted to analyze the complex structure of mAb with N-NTD, we chose three high-affinity antibodies against N-NTD to produce recombinant Fab antibodies, which are nCoV396, nCoV416, and nCoV457 here. However, we only obtained the crystal of nCoV396 and N protein. Furthermore, we analyzed the antigen recognition epitopes by ELISA and found that the recognition epitopes of nCoV416 and nCoV457 are consistent with nCoV396 (Please see the response for Reviewer #2 point 3-(2) or Supplementary table 1).

Supplementary table 1: Epitope Mapping of SARS-CoV-2 N protein antibodies by Elisa

Peptide Number	Peptide sequence (N→C)	nCOV 0396			nCOV 0454			nCOV 0457			TT0170		
NP1(1-18aa)	MSDNQPQNQRNAPRITFGGSK (Biotin)	0.021	0.016	0.013	0.022	0.022	0.029	0.049	0.038	0.035	0.011	0.011	0.015
NP2(7-24aa)	QNQRNAPRITFGGSDSTGSK (Biotin)	0.014	0.009	0.005	0.016	0.016	0.027	0.039	0.029	0.026	0.004	0.004	0.012
NP3(13-30aa)	PRITFGGSDSTGNSQNGGSK (Biotin)	0.016	0.009	0.007	0.018	0.022	0.031	0.041	0.032	0.028	0.006	0.008	0.015
NP4(19-36aa)	GFSDSTGNSQNGERSGARGSK (Biotin)	0.017	0.011	0.007	0.022	0.023	0.032	0.041	0.032	0.028	0.008	0.011	0.017
NP5(25-42aa)	GSNQNERSGARGSKRRPFGSK (Biotin)	0.019	0.013	0.008	0.023	0.022	0.029	0.037	0.03	0.028	0.009	0.012	0.015
NP6(31-48aa)	ERSGARGSKRRPQGLPNNGSK (Biotin)	0.018	0.012	0.005	0.022	0.023	0.037	0.041	0.03	0.025	0.008	0.01	0.014
NP7(37-54aa)	SKRRPQGLPNNTASWFTGSK (Biotin)	0.028	0.015	0.017	0.03	0.027	0.032	0.042	0.031	0.03	0.012	0.01	0.017
NP8(43-60aa)	QGLFNNTASWFTALTQHGSK (Biotin)	0.021	0.013	0.015	0.028	0.03	0.037	0.026	0.024	0.022	0.008	0.007	0.011
NP9(49-66aa)	TASWFTALTQHGKEDLKFGSK (Biotin)	0.023	0.017	0.016	0.027	0.039	0.038	0.034	0.026	0.023	0.007	0.012	0.014
NP10(55-72aa)	ALTHGKEDLKFPFGQVGVGSK (Biotin)	0.027	0.016	0.017	0.028	0.036	0.047	0.039	0.031	0.024	0.007	0.012	0.015
NP11(61-78aa)	KEDLKFPFGQVFPINTNSGSK (Biotin)	0.025	0.016	0.016	0.031	0.033	0.045	0.032	0.025	0.022	0.007	0.008	0.015
NP12(67-84aa)	FRGQGVFPINTNSPDDIGSK (Biotin)	0.024	0.016	0.015	0.031	0.036	0.046	0.041	0.026	0.023	0.007	0.006	0.014
NP13(73-90aa)	PINTNSPDDIGIYRAGSK (Biotin)	0.029	0.025	0.025	0.03	0.033	0.033	0.066	0.06	0.056	0.036	0.03	0.028
NP14(79-96aa)	SPDDIGIYRRARRIRGGSK (Biotin)	0.025	0.025	0.026	0.02	0.024	0.027	0.061	0.044	0.038	0.023	0.026	0.032
NP15(85-102aa)	GIYRRARRIRGGDGKMGSK (Biotin)	0.03	0.028	0.023	0.023	0.025	0.028	0.067	0.051	0.047	0.029	0.035	0.037
NP16(91-108aa)	TRRIRGGDGKMDLSPRWGSK (Biotin)	0.033	0.03	0.027	0.025	0.026	0.029	0.059	0.053	0.056	0.034	0.035	0.038
NP17(97-114aa)	GDGKMDLSPRWYFYLGGSK (Biotin)	0.026	0.034	0.031	0.024	0.028	0.031	0.065	0.056	0.061	0.04	0.034	0.044
NP18(103-120aa)	DLSPRWYFYLGTGPEAGGSK (Biotin)	0.031	0.027	0.03	0.026	0.027	0.03	0.062	0.056	0.063	0.031	0.032	0.041
NP19(109-126aa)	YFYLGTGPEAGLPGANGSK (Biotin)	0.035	0.024	0.024	0.032	0.026	0.033	0.044	0.037	0.044	0.016	0.015	0.042
NP20(115-132aa)	TGPEAGLPGANKDGIWGSK (Biotin)	0.021	0.015	0.012	0.028	0.025	0.036	0.045	0.031	0.026	0.013	0.008	0.03
NP21(121-138aa)	LPYANKDGIWVATEGAGSK (Biotin)	0.027	0.018	0.014	0.029	0.023	0.036	0.04	0.03	0.031	0.012	0.009	0.027
NP22(127-144aa)	KDGIWVATEGALNTPKDGSK (Biotin)	0.026	0.018	0.012	0.031	0.029	0.046	0.044	0.03	0.032	0.013	0.013	0.032
NP23(133-150aa)	VATEGALNTPKDHIGTRNGSK (Biotin)	0.022	0.016	0.011	0.033	0.031	0.041	0.044	0.032	0.036	0.013	0.012	0.034
NP24(139-156aa)	LNTPKDHIGTRNFANNAAGSK (Biotin)	0.021	0.011	0.01	0.031	0.033	0.049	0.047	0.039	0.033	0.013	0.014	0.031
NP25(145-162aa)	HIGTRNFANNAIVLQLPGSK (Biotin)	0.031	0.027	0.026	0.052	0.044	0.054	0.036	0.034	0.033	0.011	0.017	0.023
NP26(151-168aa)	FANNAIVLQLPQGTTLPGSK (Biotin)	0.02	0.023	0.022	0.041	0.04	0.054	0.041	0.031	0.028	0.007	0.01	0.015
NP27(157-174aa)	IVLQLPQGTTLPKGYAEGSK (Biotin)	2.439	2.02	2.34	1.686	1.79	2.03	2.193	2.142	1.648	0.007	0.011	0.018
NP28(163-180aa)	QGTTLPKGYAEGSRGGGSK (Biotin)	2.215	2.007	1.83	1.86	1.844	2.216	2.047	1.795	1.601	0.009	0.014	0.02
NP29(169-186aa)	KGYAEGSRGGGQASSRSGSK (Biotin)	0.03	0.023	0.028	0.043	0.042	0.059	0.034	0.031	0.023	0.011	0.018	0.02
NP30(175-192aa)	GSRGGGQASSRSSRNRNGSK (Biotin)	0.028	0.021	0.023	0.04	0.038	0.069	0.037	0.029	0.022	0.008	0.012	0.018
NP31(181-198aa)	QASSRSSRNRNRNRNGSK (Biotin)	0.018	0.022	0.027	0.034	0.031	0.042	0.045	0.045	0.046	0.037	0.032	0.041
NP32(187-204aa)	SSRNRNRNRNTPGSSRGSK (Biotin)	0.018	0.03	0.024	0.023	0.026	0.042	0.047	0.04	0.042	0.024	0.023	0.035
NP33(193-210aa)	SSRNRNRNRNTPGSSRGSK (Biotin)	0.021	0.028	0.023	0.024	0.027	0.043	0.051	0.048	0.049	0.028	0.024	0.036
NP34(199-216aa)	PGSSRGTSPARMAGNGDGSK (Biotin)	0.027	0.023	0.024	0.026	0.027	0.043	0.054	0.052	0.058	0.033	0.029	0.047
NP35(205-222aa)	TSPARMAGNGDAAALALLGSK (Biotin)	0.026	0.03	0.029	0.029	0.03	0.044	0.054	0.048	0.053	0.035	0.029	0.045
NP36(211-228aa)	ACNGGDAALALLLDRLNGSK (Biotin)	0.02	0.031	0.031	0.025	0.032	0.039	0.051	0.048	0.053	0.041	0.03	0.045
NP37(217-234aa)	AALALLLDRLNQLSKMGSK (Biotin)	0.015	0.013	0.015	0.032	0.03	0.043	0.045	0.034	0.029	0.01	0.012	0.015
NP38(223-240aa)	LLDRLNQLSKMSGKQGGSK (Biotin)	0.012	0.012	0.01	0.026	0.025	0.039	0.036	0.031	0.024	0.006	0.005	0.012
NP39(229-246aa)	QLSKMSGKQGGQGGTVGSK (Biotin)	0.019	0.013	0.012	0.026	0.029	0.035	0.038	0.032	0.028	0.011	0.011	0.016
NP40(235-252aa)	SGKQGGQGGTVTKKSAAGSK (Biotin)	0.019	0.014	0.013	0.032	0.034	0.041	0.041	0.034	0.028	0.013	0.013	0.017
NP41(241-258aa)	QQGQTVTKKSAAEASKKPGSK (Biotin)	0.024	0.016	0.013	0.033	0.035	0.044	0.041	0.036	0.028	0.012	0.011	0.015
NP42(247-264aa)	TKKSAEASKKPRKRTAGSK (Biotin)	0.017	0.013	0.012	0.028	0.029	0.033	0.045	0.036	0.026	0.01	0.01	0.013
NP43(253-270aa)	EASKKPRKRTATKAYNVGSK (Biotin)	0.02	0.016	0.013	0.038	0.038	0.041	0.056	0.044	0.048	0.023	0.022	0.029
NP44(259-276aa)	RKRTATKAYNVTQAFGRGSK (Biotin)	0.019	0.01	0.007	0.03	0.023	0.044	0.062	0.036	0.032	0.012	0.012	0.025
NP45(265-282aa)	TEAYNVTQAFGRRGPEQTGSK (Biotin)	0.022	0.012	0.008	0.035	0.033	0.041	0.048	0.039	0.034	0.01	0.011	0.022
NP46(271-288aa)	TQAFGRRGPEQTQGNFGDGSK (Biotin)	0.022	0.009	0.01	0.035	0.036	0.035	0.052	0.039	0.034	0.011	0.011	0.022

NP47(277-294aa)	RGPEQTQGNFGDQELIRQGSX(Biotin)	0.018	0.013	0.01	0.044	0.038	0.04	0.05	0.039	0.035	0.013	0.012	0.022
NP48(283-300aa)	QGNFGDQELIRQCTDYKHGSX(Biotin)	0.016	0.012	0.009	0.041	0.035	0.039	0.058	0.04	0.037	0.015	0.014	0.024
NP49(289-306aa)	QELIRQCTDYKHWPQIAQGSX(Biotin)	0.016	0.012	0.017	0.023	0.028	0.037	0.061	0.048	0.041	0.021	0.026	0.02
NP50(295-32aa)	GTDYKHWPQIAQFAPSASGSX(Biotin)	0.015	0.009	0.012	0.023	0.024	0.039	0.062	0.038	0.033	0.021	0.018	0.024
NP51(301-318aa)	WPQIAQFAPSASAFFGMSGX(Biotin)	0.015	0.014	0.026	0.024	0.025	0.039	0.066	0.043	0.061	0.025	0.027	0.031
NP52(301-324aa)	FAPSASAFFGMSRIGMEVGSX(Biotin)	0.015	0.02	0.022	0.027	0.029	0.041	0.061	0.045	0.054	0.031	0.027	0.024
NP53(313-330aa)	AFFGMSRIGMEVTPSGTWGSX(Biotin)	0.017	0.016	0.023	0.025	0.027	0.031	0.06	0.04	0.051	0.029	0.023	0.027
NP54(319-336aa)	RIGMEVTPSGTWLTYTAAGSX(Biotin)	0.016	0.014	0.017	0.024	0.026	0.035	0.048	0.049	0.044	0.03	0.024	0.036
NP55(325-343aa)	TPSGTWLTYTAAIKLDDKGSX(Biotin)	0.037	0.022	0.023	0.033	0.028	0.033	0.042	0.04	0.04	0.023	0.018	0.023
NP56(331-348aa)	LTYTAAIKLDDKDPNFKDGSX(Biotin)	0.03	0.012	0.011	0.028	0.023	0.036	0.038	0.036	0.032	0.019	0.016	0.03
NP57(337-354aa)	IKLDDKDPNFKDQVILLNGSX(Biotin)	0.033	0.017	0.01	0.028	0.028	0.036	0.045	0.042	0.035	0.019	0.018	0.03
NP58(343-360aa)	DPNFKDQVILLNKHIDAYGSX(Biotin)	0.031	0.017	0.012	0.032	0.029	0.039	0.047	0.043	0.036	0.021	0.019	0.032
NP59(349-366aa)	QVILLNKHIDAYKTFPPTGSX(Biotin)	0.03	0.013	0.009	0.034	0.031	0.04	0.043	0.039	0.04	0.021	0.02	0.035
NP60(355-372aa)	KHIDAYKTFPPTFPKKDKGSX(Biotin)	0.032	0.01	0.006	0.033	0.03	0.04	0.04	0.04	0.038	0.021	0.02	0.036
NP61(361-378aa)	KTFPPTFPKKDKKKKADKGSX(Biotin)	0.034	0.021	0.022	0.041	0.038	0.044	0.038	0.032	0.034	0.011	0.016	0.023
NP62(367-384aa)	EPKKDKKKKADKQALPQGSX(Biotin)	0.024	0.015	0.013	0.034	0.038	0.041	0.04	0.027	0.027	0.008	0.007	0.018
NP63(373-390aa)	KKKADKQALPQRKQKQGSX(Biotin)	0.02	0.015	0.014	0.039	0.04	0.053	0.035	0.023	0.025	0.004	0.007	0.021
NP64(379-396aa)	TQALPQRKQKQTVTLLEGSX(Biotin)	0.021	0.015	0.013	0.041	0.042	0.053	0.037	0.024	0.022	0.008	0.006	0.018
NP65(385-402aa)	RQKRQQTVTLLEPAADLDDGSX(Biotin)	0.026	0.018	0.014	0.036	0.037	0.047	0.036	0.023	0.021	0.008	0.008	0.018
NP66(391-408aa)	TVTLLEPAADLDDFSKQLQGSX(Biotin)	0.02	0.015	0.011	0.038	0.043	0.06	0.045	0.026	0.025	0.007	0.009	0.021
NP67(397-414aa)	AADLDDFSKQLQSMSSAGSX(Biotin)	0.02	0.014	0.009	0.032	0.028	0.04	0.046	0.031	0.029	0.01	0.01	0.012
NP68(403-419aa)	FSKQLQSMSSADSTQAAGSX(Biotin)	0.016	0.01	0.007	0.027	0.028	0.031	0.039	0.03	0.029	0.007	0.007	0.011

Reactivity of the antibodies against synthetic peptides measured by ELISA.
The highlighted part indicates that the antibody binds to this peptide.

To test whether the two other antibodies have similar functions with nCoV396, we next verified the function of nCoV457 in ex vivo assays. Consistently, nCoV457 has a similar but weaker inhibitory effect to nCoV396, which may be due to nCoV457 and N protein's weaker affinity than nCoV396 (**Supplementary Fig. 3a, b**).

Supplementary Figure 3 | Other antibodies (nCoV454 and nCoV457) compromise SARS-CoV-2 and SARS-CoV N-induced complement hyperactivation (a) The Michealis-Menten curve of *ex vivo* SARS-CoV-2 N protein-induced excessive cleavage of C2 in the serum. The v_0 is indicated on the Y-axis, the concentration of substrate is indicated on the X-axis. mAb nCoV396 (dark blue), nCoV457 (cyan), nCoV454 (black), Negative control with other protein (ENL) expressed in *E. coli* (orange), and Blank control (blue) are presenting. **(b)** the corresponding kinetics parameters of **(a)** are presented. **(c)** The Michealis-Menten curve of *ex vivo* SARS-CoV N protein-induced excessive cleavage of C2 in the serum, with similar representation as **(a)**. **(d)** The corresponding kinetics parameters of **(c)** are presented.

MINOR COMMENTS:

1. Page 4 - the authors state "a recent retrospective observational study of ...revealed that complement disorder was associated with "....Please be more specific regarding this observation.

Response: We concur with the reviewer's suggestion and add the following sentence at Line 58-65: 'Patients with age-related macular degeneration (AMD, a proxy for complement activation disorders) were at significantly increased risk of adverse clinical outcomes following SARS-CoV-2 infection. Conversely, patients with complement deficiency disorders genetic background required little mechanical respiration or succumbed to their illness. Together, these data suggest that hyperactive complement predispose individuals to adverse outcomes associated with SARS-CoV-2 infection.'

2. Page 4 - A recent preprint study....it is good that the authors have acknowledged this study is a preprint study. It is somewhat concerning this study was uploaded on a preprint server in March 2020 but has yet not been published, although is widely cited. To that extent, at the end of this paragraph, the authors should add a sentence about how the current study can help further advance the role of N-protein inducing complement activation. They should also subsequently address this in detail in the discussion, possibly via focusing on the RNA binding domain of the N-protein. Do the authors suggest that this domain is what is contributing to complement-mediated hyper activation? Or how do they put the findings of their current study, into perspective with the two prior studies (references 19 and 21)?

*Response: We have supplemented the experiment that SARS-CoV-2 N protein can directly improve the enzymatic activity efficiency of the catalytic domain of MASP-2 purified in vitro (**Supplementary Fig S4**). These results complement the *ex vivo* data in the article, and together they can prove that N protein is contributing to complement-mediated hyperactivation. From our perspective, we found that N protein directly interacted with MASP-2, induced the activity of MASP-2, and served as a potent target for antibody therapy of COVID-19.*

3. The authors mention that ZD004 and ZD006 had only minimal levels of antibody response to the S protein, but much higher titers to the N protein. Hence, how likely is the current schema proposed valid in the human population? The authors may want to consider incorporating references such as McAndrews et al. (<https://insight.jci.org/articles/view/142386>), among others, into their Discussion to address this issue.

Response: Asmaa Hachim et al. have reported that antibodies to N protein developed earlier than S protein-specific antibodies.⁸ Similarly, Baoqing Sun et al. found that N-IgG was significantly higher in ICU patients than in non-ICU patients.⁹ Consistently with their results, our study found several COVID-19 patients infected with SARS-CoV-2 have higher antibody titers to N protein than S protein. Due to the small sample size, it cannot be said that this is a common phenomenon in the human population. We concur with the reviewer's suggestion and have edited the sentence at Line 321-324.

4. Regarding the antibody repertoires, how did the authors ensure that there was no cross-reactivity of monoclonal Abs to other viral proteins? Additionally, among 32 mAbs that bound to N-FL, 13 Abs bound to N-NTD and one Ab bound to N-CTD, but which part of the protein did the remaining 18 Abs bind to?

Response: To test whether the antibody nCoV396 or other mAbs have the potential to cross-react with other highly pathogenic coronavirus, we perform the ex vivo and in vitro assays in the context of another relative N protein. As shown in **Supplementary Fig. 3c, d**, mAbs have the potential to cross-react with SARS-CoV N protein. For the other 18 Abs binding, the reviewer raises an important point. However, it is labor intense, time consuming, unpredictably to express the rest of the nucleocapsid portion separately due to the nature of the protein. These regions belong to disorder or flexible parts of the protein, although we have worked on these several times. The comprehensive studies have suggested that the nucleocapsid compact functional domains are its NTD and CTD. These two domains play several vital roles in viral RNA recognition, viral genomic RNA packing, high-order structure formation of viral ribonucleoproteins (RNP), etc. Therefore, we focus our subsequent studies on the monoclonal antibodies that bind to N-NTD, N-CTD or full-length protein in this project. Nevertheless, we agree with the reviewer that it is worth checking whether these monoclonal antibodies are bound to other regions of the nucleocapsid protein in the future work once we can express these flexible regions.

Reference

- 1 Grosseohme, N. E. *et al.* Coronavirus N protein N-terminal domain (NTD) specifically binds the transcriptional regulatory sequence (TRS) and melts TRS-cTRS RNA duplexes. *J Mol Biol* **394**, 544-557 (2009).
- 2 Lin, S. Y. *et al.* Structural basis for the identification of the N-terminal domain of coronavirus nucleocapsid protein as an antiviral target. *J Med Chem* **57**, 2247-2257 (2014).
- 3 Chen, C. Y. *et al.* Structure of the SARS coronavirus nucleocapsid protein RNA-binding dimerization domain suggests a mechanism for helical packaging of viral RNA. *J Mol Biol* **368**

(2007).

- 4 Yao, H. *et al.* Molecular Architecture of the SARS-CoV-2 Virus. *Cell* **183**, 730-738 e713 (2020).
- 5 Robert, X. & Gouet, P. Deciphering key features in protein structures with the new ENDscript server. *Nucleic Acids Res* **42**, W320-324 (2014).
- 6 Hung, H. C. *et al.* Development of an anti-influenza drug screening assay targeting nucleoproteins with tryptophan fluorescence quenching. *Anal Chem* **84**, 6391-6399 (2012).
- 7 Rambaldi, A. *et al.* Endothelial injury and thrombotic microangiopathy in COVID-19: Treatment with the lectin-pathway inhibitor narsoplimab. *Immunobiology* **225**, 152001 (2020).
- 8 Hachim, A. *et al.* ORF8 and ORF3b antibodies are accurate serological markers of early and late SARS-CoV-2 infection. *Nat Immunol* **21**, 1293-1301 (2020).
- 9 Sun, B. *et al.* Kinetics of SARS-CoV-2 specific IgM and IgG responses in COVID-19 patients. *Emerg Microbes Infect* **9**, 940-948 (2020).

REVIEWER COMMENTS

Reviewer #1 (Remarks to the Author):

I understand the difficulty to characterize the other monoclonal antibodies, which recognize SARS-CoV-2 N protein but don't bind to N-NTD or N-CTD.
The authors adequately addressed the raised comments and questions and improved the manuscript.

Reviewer #2 (Remarks to the Author):

The authors have addressed the concerns raised and the improvements suggested in the review and the edited manuscript is now suitable for publication.

Reviewer #3 (Remarks to the Author):

The authors have provided appropriate responses to most of the comments that I had raised in my original review. The addition of Supplementary Figure 4 is helpful to understand their system, designed to probe MASP-2 mediated C2 cleavage. Addition of text regarding nCoV457 is also helpful to understand why nCoV396 was selected, as is a clarification regarding the patients with autoimmune disease. The comments below are primarily to help the scientific community better understand the manuscript, and to facilitate reproducibility -

MINOR COMMENTS

1. The methodology described in response to Reviewer 3, question 1 needs to be incorporated into the Methods (either main manuscript or Supplement). If it has already been reported in this level of detail included in the response to reviewer comments, I apologize, as I was not able to find it. In this situation, please do point me to where it is in the manuscript. Having the level of detail that has been provided in the response to the reviewer would be necessary to incorporate in the manuscript to ensure reproducibility by the scientific community.
2. Reviewer 3 question 3 - I am confused about the answer and the inclusion of patient samples in different experiments is still somewhat hard to follow. It appears the authors are referring to Extended Table 6, instead of Extended Table 5, as reported in their response. Additionally, I see 10 patients in this table, not 6. It would be helpful if the authors clarify in the Table itself which samples were actually used. Moreover, while Supplementary Figure 6 is helpful, this includes only one patient and control. They should include at least n=3 in both groups, ideally more. A minor change is that the legend of Figure 4 needs to be updated to 'patients with abnormal C3 or C4 values'. To that extent, the last figure in Figure 4E (bottom right) is confusing; it does not include the patients from 4b and 4d; nor a healthy control. If this could be clarified why it was done this way, or ideally, the data provided, that would be helpful.
3. Reviewer 3 question 4 - While nCoV457 was compared to nCoV396, why was the same thing not done for nCoV416? It appears nCoV454 was tested, which makes this response somewhat confusing.
4. While the authors have added the references of Hachim et al and Sun et al in lines 321-4, they should also add the sentence "due to the small sample size..." (that they wrote in their response) to the main manuscript, and cite the McAndrews manuscript, to demonstrate some equipoise.
5. The answer to Reviewer 3, Minor Comment#4 is satisfactory, but needs to be included in the main manuscript under a paragraph on limitations.

REVIEWERS' COMMENTS

Reviewer #3 (Remarks to the Author):

The authors have provided appropriate responses to most of the comments that I had raised in my original review. The addition of Supplementary Figure 4 is helpful to understand their system, designed to probe MASP-2 mediated C2 cleavage. Addition of text regarding nCoV457 is also helpful to understand why nCoV396 was selected, as is a clarification regarding the patients with autoimmune disease. The comments below are primarily to help the scientific community better understand the manuscript, and to facilitate reproducibility.

Response:

We thank reviewer's opinions and accept his/her suggestions which key clarifications need addressing in the revised version of the manuscript. The appropriate sentences have been edited from the revised version.

MINOR COMMENTS

1. The methodology described in response to Reviewer 3, question 1 needs to be incorporated into the Methods (either main manuscript or Supplement). If it has already been reported in this level of detail included in the response to reviewer comments, I apologize, as I was not able to find it. In this situation, please do point me to where it is in the manuscript. Having the level of detail that has been provided in the response to the reviewer would be necessary to incorporate in the manuscript to ensure reproducibility by the scientific community.

Response:

We concur with the reviewer's recommendation that the response to Reviewer 3 - Question 1 should be incorporated into the Methods. For clarifying, we have added the methodology description in the line 494-500 of the Methods part.

2. Reviewer 3 question 3 - I am confused about the answer and the inclusion of patient samples in different experiments is still somewhat hard to follow. It appears the authors are referring to Extended Table 6, instead of Extended Table 5, as reported in their response. Additionally, I see 10 patients in this table, not 6. It would be helpful if the authors clarify in the Table itself which samples were actually used. Moreover, while Supplementary Figure 6 is helpful, this includes only one patient and control. They should include at least n=3 in both groups, ideally more. A minor change is that the legend of Figure 4 needs to be updated to 'patients with abnormal C3 or C4 values'. To that extent, the last figure in Figure 4E (bottom right) is confusing; it does not include the patients from 4b and 4d; nor a healthy control. If this could be clarified why it was done this way, or ideally, the data provided, that would be helpful.

Response:

We must apologize for our mistakes at referring a wrong table. A corrected sentence has been edited from the revised version.

We understood the confusing nomenclature of Patient ID in the main text, more likely due to the discontinuous nomenclature. In order to describe the content of our experiment more clearly, we

have renumbered the sample with the continuous nomenclature shared the same prefix ‘Serum’. In the new sample nomenclature, Serum-01 to Serum-03 are samples of health donors, while Serum - 04 to Serum-13 are serum samples with abnormal serologic C3 which are indicators of abnormal complement system in clinical. We have summarized the relevant information of all the serum samples into the Supplementary Table 6. Their applications in the assays as shown below:

Sample ID	Patient ID	C3 (0.7-1.4 g/L) ^(a)	C4 (0.1-0.4 g/L) ^(a)	Application
Serum-01	Health-110	1.00	0.22	Fig. 4b Supplementary Fig. 5a
Serum-02	Health-113	0.84	0.25	Fig. 4b Supplementary Fig. 5a
Serum-03	Health-117	1.14	0.33	Fig. 4b Supplementary Fig. 5a
Serum-04	Patient-81	1.48	0.21	Fig. 4b Supplementary Fig. 5a
Serum-05	Patient-123	1.55	0.26	Fig. 4b Supplementary Fig. 5a
Serum-06	Patient-130	1.66	0.23	Fig. 4b Supplementary Fig. 5a
Serum-07	Patient-49	1.43	0.26	Fig. 4d
Serum-08	Patient-20	1.57	0.37	Fig. 4f, g
Serum-09	Patient-19	1.47	0.23	Fig. 4g Supplementary Fig. 5b
Serum-10	Patient-34	1.43	0.37	Fig. 4g Supplementary Fig. 5b
Serum-11	Patient-38	1.57	0.28	Fig. 4g Supplementary Fig. 5b
Serum-12	Patient-71	1.58	0.31	Fig. 4g, Supplementary Fig. 7a
Serum-13	Patient-72	1.48	0.28	Supplementary Fig. 7c

a Reference value range for healthy person. All serum samples were detected by Clinical Laboratory Group of Department of Experimental Medicine of The Fifth Affiliated Hospital, Sun Yat-sen University.

- (1) To clarify the results in Fig.4b and Supplementary Fig. 5a, Serum-01 to -06 are used to compare the MASP-2 activity to C2 of serum sample with normal C3 values (Health-110/113/117, n=3) and serum sample with abnormal C3 values (Patient-81/123/130, n=3).
- (2) Serum-07 is used to prove that different concentration SARS-CoV-2 N protein can promote the enzymatic activity of MASP2 toward C2 substrate, corresponding to Fig. 4d result.

- (3) Serum-08 is used to prove that different concentration SARS-CoV-2 N protein antibody (nCoV396) can inhibit the enzymatic activity of MASP-2 induced by N protein, corresponding to Fig. 4f result.
- (4) Serum-09 to -11 are used to expand of the sample size to prove that SARS-CoV-2 N protein antibody (nCoV396) can inhibit the ability of MASP-2 to cleave C2 induced by N protein, corresponding to Fig. 4g result.
- (5) Serum-12 is used to prove that antibodies (nCoV396, nCoV454 and nCoV457) compromise SARS-CoV-2 N-induced complement hyperactivation, corresponding to Supplementary Fig. 7a, b result.
- (6) Serum-13 is used to prove that antibodies (nCoV396, nCoV454 and nCoV457) compromise SARS-CoV N-induced complement hyperactivation, corresponding to Supplementary Fig. 7c, d result.

In addition, to clarify the independent experimental number with reviewer's critique, we now have added three health control vs patients paired data in Supplementary Fig. 5a (the previous version Supplementary Fig. 6). The results are consistent with the previous conclusions, the serum MASP-2 activity to C2 of the healthy donors are much weaker than those from abnormal serologic samples.

Supplementary Figure 5 (a) Data are from three donors with abnormal serologic C3 values (patient) and three donors with normal serologic C3 values (health). The Michaelis-Menten curve shows the effect of the N protein in the former was higher than latter on the substrate C2 cleavage of MASP-2. **(b)** The mAb nCoV396 inhibits the N protein-induced excessive cleavage of C2 in the serum with abnormal serologic C3 values. Negative ctrl (orange curve) represents reactions containing other protein (ENL) expressed in *E. coli*. instead of SARS-CoV-2 N protein, and blank ctrl (blue curve) without SARS-CoV-2 N protein. All samples were performed in triplicates and mean were presented.

4. While the authors have added the references of Hachim et al and Sun et al in lines 321-4, they should also add the sentence "due to the small sample size..." (that they wrote in their response) to the main manuscript, and cite the McAndrews manuscript, to demonstrate some equipoise.

Response:

We are very grateful for your suggestions. We have added the corresponding sentence in the line 312-315 and cited the McAndrews literature into the latest manuscript.

5. The answer to Reviewer 3, Minor Comment#4 is satisfactory, but needs to be included in the main manuscript under a paragraph on limitations.

Response:

We are very willing to accept your pertinent suggestions, in the discussion part of the article we add the following content in the line 368- 377: "For the other 18 antibodies, however, it is labor intense, time consuming, unpredictably to express the rest of the nucleocapsid portion separately due to the nature of the protein. These regions belong to disorder or flexible parts of the protein, although we have worked on these several times. The comprehensive studies have suggested that the nucleocapsid compact functional domains are its NTD and CTD. These two domains play several vital roles in viral RNA recognition, viral genomic RNA packing, high-order structure formation of viral ribonucleoproteins (RNP), etc. Therefore, we focus our subsequent studies on the monoclonal antibodies that bind to N-NTD, N-CTD or full-length protein in this project. Nevertheless, we believe that it is worth checking whether these monoclonal antibodies are bound to other regions of the nucleocapsid protein in the future work."